# Training Diffusion Classifiers with Denoising Assistance

## Abstract

Score-matching and diffusion models have emerged as state-of-the-art generative models for both conditional and unconditional generation. Classifier-guided diffusion models are created by training a classifier on samples obtained from the forward-diffusion process (i.e., from data to noise). In this paper, we propose denoising-assisted (DA) classifiers wherein the diffusion classifier is trained using both noisy and denoised examples as simultaneous inputs to the model. We differentiate between denoising-assisted (DA) classifiers and noisy classifiers, which are diffusion classifiers that are only trained on noisy examples. Our experiments on Cifar10 and Imagenet show that DA-classifiers improve over noisy classifiers both quantitatively in terms of generalization to test data and qualitatively in terms of perceptually-aligned classifier-gradients and generative modeling metrics. We theoretically characterize the gradients of DA-classifiers to explain improved perceptual alignment. Building upon the observed generalization benefits of DA-classifiers, we propose and evaluate a semi-supervised framework for training diffusion classifiers and demonstrate improved generalization of DA-classifiers over noisy classifiers.

## 1 Introduction

Score models are unnormalized probabilistic models that model the probability density in terms of its score function – that is, the gradient of the log-likelihood. Score models are advantageous as compared to contemporary generative models such as GANs, VAEs, Autoregressive models and normalizing flows as they require neither adversarial optimization nor restricted architecture families, while achieving state-of-the-art performance across multiple modalities. Score models can be trained using any score-matching objective such as the implicit score-matching (Hyvärinen, 2005), the sliced score-matching (Song et al., 2020b) or the denoising score-matching (Vincent, 2011) methods. Song et al. (2021) use denoising score-matching to unify Multi-scale Score Matching (Song and Ermon, 2019; 2020) and Denoising Diffusion Probabilistic Models (Ho et al., 2020) under a stochastic diffusion framework. We briefly review the stochastic diffusion score-matching framework introduced by Song et al. (2021) in Section 2.

In this work, we focus on supervised and semi-supervised training of classifier-guided diffusion models that utilize the gradients of classifier with respect to its input to guide the sampling by maximizing the log-probability of a target class. To this end, a classifier is trained on noisy images sampled from the forward diffusion as described in Section 2; the forward diffusion is usually described such that the distribution of examples at a given diffusion time can be specified as a closed-form Gaussian implying that the examples sampled from the diffusion contain Gaussian noise. We use the term diffusion classifiers to refer to classifiers trained on diffused examples.

We systematically expose deficiencies of noisy classifiers in terms of (a) generalization to unseen examples in fully-supervised and semi-supervised settings, (b) perceptual alignment of classifier-gradients, and (c) conditional image-generation. Motivated to address these deficiencies, we introduce a simple modification to the training of the classifier: instead of providing only the noisy image to the classifier, we give both the noisy image *and also* its corresponding denoised image as simultaneous inputs to the classifier; we refer to this as a denoising-assisted (DA) classifier. To train the DA-classifier, we reuse the same denoising models used to parameterize the diffusion model. The benefits of learning to classify denoised examples is not limited to simplifying the learning task; interestingly, our experiments show that this simple modification results in improved generalization to unseen

examples and improved perceptual alignment of classifier gradients. We support these empirical observations, including the perceptual gradient alignment, with a theoretical analysis. Collectively, the improvements in generalization and classifier-gradients lead to improved image generation.

Finally, we propose and evaluate a framework for semi-supervised training of classifier-guided conditional score-models that builds upon the generalization abilities of DA-classifiers in label-limited settings wherein the training data consists of a largely unlabelled dataset, along with a relatively small number of labelled examples.

**Contributions.** In summary:
(1) We introduce denoising-assisted (DA) classifiers. We analyze DA-classifiers empirically with Imagenet and CIFAR10 datasets, showing improvements over noisy diffusion classifiers.
(2) We include analytical discussions that support the above observations.
(3) We introduce a semi-supervised framework for training diffusion classifiers, inspired by state-of-the-art semi-supervised methods and show that denoising-assisted classifiers generalize better in label-limited settings.

## 2 BACKGROUND

Score models are probabilistic models of the data that enable sampling and exact inference of log-likelihoods. Song et al. (2021) propose a framework generalizing Multi-scale score matching (Song and Ermon, 2019; 2020) and Denoising Diffusion Probabilistic Models (Ho et al., 2020). Concretely, the framework consists of two components: 1) the forward-diffusion (i.e., data to noise) stochastic process, and 2) a learnable score-function that can then be used for the reverse-diffusion (i.e., noise to data) stochastic process.

The forward diffusion stochastic process $\{\mathbf{x}_t\}_{t \in [0,T]}$ starts at data, $\mathbf{x}_0$, and ends at noise, $\mathbf{x}_T$. We let $p_t(\mathbf{x})$ denote the probability density of $\mathbf{x}$ at time $t$, so, e.g., $p_0(\mathbf{x})$ is the distribution of the data, and $p_T(\mathbf{x})$ is the distribution of the noise. The diffusion is structured so that $p_T(\mathbf{x})$ is independent of the starting point at $t = 0$. This process is defined with a stochastic-differential-equation (SDE):

$$d\mathbf{x} = \mathbf{f}(\mathbf{x}, t)\, dt + g(t)\, d\mathbf{w}, \tag{1}$$

where $\mathbf{w}$ denotes a standard Wiener process, $\mathbf{f}(\mathbf{x}_t, t)$ is a drift coefficient, and $g(t)$ is a diffusion coefficient. The drift and diffusion coefficients are usually manually specified without learnable parameters; this lets us obtain closed-form solutions to the forward-diffusion SDE. For example, if $\mathbf{f}$ is linear in $\mathbf{x}$, the solution to the SDE is a time-varying gaussian distribution whose mean $\mu(\mathbf{x}_0, t)$ and standard deviation $\sigma(t)$ can be exactly computed. We use $p_t(\mathbf{x}|\mathbf{x}_0)$ to denote the probability density function of $\mathbf{x}_t$ when the diffusion is seeded at $\mathbf{x}_0$.

To sample from $p_0(\mathbf{x})$ starting with samples from $p_T(\mathbf{x})$, we have to solve the reverse diffusion SDE (Anderson, 1982):

$$d\mathbf{x} = [\mathbf{f}(\mathbf{x}, t) - g(t)^2 \nabla_{\mathbf{x}} \log p_t(\mathbf{x})]\, dt + g(t)\, d\bar{\mathbf{w}}, \tag{2}$$

where $d\bar{\mathbf{w}}$ is a standard Wiener process when time flows from T to 0, and $dt$ is an infinitesimal negative timestep. In practice, the score function $\nabla_{\mathbf{x}} \log p_t(\mathbf{x})$ is estimated by a neural network $s_\theta(\mathbf{x}, t)$, parameterized by $\theta$, trained to optimize the following score-matching loss:

$$\int_0^T \mathbb{E}_{\mathbf{x} \sim p_t(\mathbf{x})}[\lambda(t)||\nabla_{\mathbf{x}} \log p_t(\mathbf{x}) - \mathbf{s}_\theta(\mathbf{x}, t)||_2^2]dt \tag{3}$$

where $\lambda(t)$ is a positive real number introduced to balance out the score-matching objective across various time steps. Using samples from the training dataset, we can define an empirical density function for $t = 0$ as $p_0(\mathbf{x}) = \frac{1}{N} \sum_{\mathbf{x}_i \in p_{data}} \delta(\mathbf{x} - \mathbf{x}_i)$ and then obtain samples from $p_t(\mathbf{x})$ by first sampling $\mathbf{x}_0 \sim p_0$ and then solving the forward-diffusion SDE (Eq. 1). If the solution to the SDE is a Gaussian distribution whose mean $\mu(\mathbf{x}_0, t)$ and covariance matrix $\Sigma(t)$ can be determined in a closed-form, we can empirically define $p_t(\mathbf{x})$ as a mixture of $N$ gaussians; for such SDE's, we can also estimate the score-function $\nabla_{\mathbf{x}} \log p_t(\mathbf{x})$ in the closed-form for evaluating the score-matching loss: this is usually referred to as denoising score matching as the score-function points in the denoising direction – i.e., towards $\mu(\mathbf{x}_0, t)$.

**Class-Conditional Score-based SDE Models**  Given a data distribution whose samples can be classified into $C$ classes, class-conditional score-models are trained to estimate $\nabla_{\mathbf{x}} \log p(\mathbf{x}, t|y)$ where $y \in [1, C]$ is the class label. Classifier-free conditional models directly learn $s_\theta(\mathbf{x}, t|y)$ by taking $y$ as an additional input. On the other hand, *classifier-guided* conditional models learn the probability distribution $p(y|\mathbf{x}, t)$ using a classifier and then combine this with the learnt unconditional score (i.e., $s_\theta(\mathbf{x}, t)$) using Bayes rule: $p(\mathbf{x}, t|y) = \frac{p(y|\mathbf{x},t)p(\mathbf{x},t)}{p(y)}$: applying log on both sides and taking the derivative with respect to $\mathbf{x}$, we get

$$s_\Theta(\mathbf{x}, t|y) = \nabla_{\mathbf{x}} \log p_\phi(y|\mathbf{x}, t) + s_\theta(\mathbf{x}, t) \tag{4}$$

where $\phi$ denotes the parameters of the classifier, so the full model is parameterized by $\Theta = \{\theta, \phi\}$. In this work, we use classifier-guided score models as this allows us to reuse pre-trained larger score-models while training smaller classifier-models: for example, Song et al. (2021) generate class-conditional CIFAR10 examples using a score-model having 107M parameters and a classifier-model having 1.5M parameters. Classifier-guidance also allows us to flexibly incorporate new definitions of classes without requiring to retrain the larger score-model.

Song et al. (2021) suggest a simple sum of the cross-entropy losses over examples sampled from the diffusion for training the classifier $p_\phi$:

$$\mathcal{L}_{\text{CE}} = \mathbb{E}_{t,\mathbf{x}}[-\log p_\phi(y|\mathbf{x}, t)] \tag{5}$$

where, $t \sim \mathcal{U}(0, T)$, $(\mathbf{x}_0, y) \sim p_0(\mathbf{x})$ and $\mathbf{x} \sim p_t(\mathbf{x}|\mathbf{x}_0)$. Dhariwal and Nichol (2021a) introduce a classifier gradient scale $\lambda_s$ — in which, the classifier gradient in Eq. 4 is scaled with $\lambda_s$ — as a control for trading off fidelity vs diversity. Higher scale corresponds to higher fidelity and lower diversity.

## 3 Denoising-Assisted (DA) Classifier

Given samples $(\mathbf{x}_0, y) \sim p_0(\mathbf{x})$ and a forward diffusion SDE, our goal is to train a classifier on examples sampled from the forward diffusion. Suppose that the solution to the SDE at time $t$, when seeded with $\mathbf{x}_0$ at time 0, can be specified as a Gaussian distribution with mean $\mu(\mathbf{x}_0, t)$ and diagonal covariance $\sigma^2(t)\mathbf{I}$. We write this as $p_t(\mathbf{x}|\mathbf{x}_0) = \mathcal{N}(\mathbf{x} \mid \mu(\mathbf{x}_0, t), \sigma^2(t)\mathbf{I})$. For training the DA-classifier, rather than using the typical (noisy) input, we propose to use as input both $\mathbf{x} \sim p_t(\mathbf{x}|\mathbf{x}_0)$ *and* a denoised image $\hat{\mathbf{x}}$:

$$\mathcal{L}_{\text{CE}} = \mathbb{E}_{t,\mathbf{x}}[-\log p_\phi(y|\mathbf{x}, \hat{\mathbf{x}}, t)] \tag{6}$$

where, $t \sim \mathcal{U}(0, T)$, $(\mathbf{x}_0, y) \sim p_0(\mathbf{x})$ and $\mathbf{x} \sim p_t(\mathbf{x}|\mathbf{x}_0)$. We compute $\hat{\mathbf{x}}$ using a pretrained score network, $s_\theta$, as $\hat{\mathbf{x}} = \mathbf{x} + \sigma^2(t)s_\theta(\mathbf{x}, t)$. Note that this denoised image refers to the estimated mean of the Gaussian that $\mathbf{x}$ was sampled from. In particular, the mean does not change with diffusion time $t$ in Variance-exploding SDEs while the mean decays to zero with diffusion time for DDPMs. We consider using a denoised example obtained at time $t$ as an additional input to the classifier for estimating $s_\Theta(\mathbf{x}, t|y)$ (Eq. 4) and this does not require any modification to the sampling algorithm. In the following, we evaluate—both quantitatively and qualitatively—the improvements obtained by using $\hat{\mathbf{x}}$ as an additional input to the classifier.

**Experiment setup.** We conduct our experiments on CIFAR10 and 256×256 Imagenet in the VE-Diffusion (Song et al., 2021) and Improved-DDPM (Nichol and Dhariwal, 2021; Dhariwal and Nichol, 2021b) settings respectively. We use pretrained score-networks for these experiments: specifically, we use the deep NCSN++ (continuous) model released by Song et al. (2021) as the score-network for CIFAR10 and the unconditional model open-sourced by Dhariwal and Nichol (2021b) as the score-network for Imagenet. We use the pretrained noisy Imagenet classifier released by Dhariwal and Nichol (2021b) while we trained the noisy CIFAR10 classifier ourselves; the Imagenet classifier is the downsampling half of the UNET with attention pooling classifier-head while we use WideResNet-28-2 as the architecture for CIFAR10. For the DA-classifier, we simply add an extra convolution that can process the denoised input: for Imagenet, we finetune the pretrained noisy classifier by adding an additional input-convolution module while we train the denoising-assisted CIFAR10 classifier from scratch. The details of the optimization are as follows: (1) for Imagenet, we fine-tune the entire network along with the new convolution-module (initialized with very small weights) using AdamW optimizer with a learning-rate of 1e-5 and a weight-decay of 0.05 for 50k steps with a batch size of 128. (2) For CIFAR10, we train both noisy and DA-classifiers for 150k steps with a batch size of

Table 1: Summary of Test Accuracies for CIFAR10 and Imagenet: each test example is diffused to a random uniformly sampled diffusion time. Both classifiers are shown the same diffused example.

| Method \ Dataset | CIFAR10 | Imagenet | |
| --- | --- | --- | --- |
| | | Top-1 | Top-5 |
| Noisy Classifier | 54.79 | 33.78 | 49.86 |
| Denoising Assisted Classifier (Ours) | **57.16** | **36.11** | **52.34** |

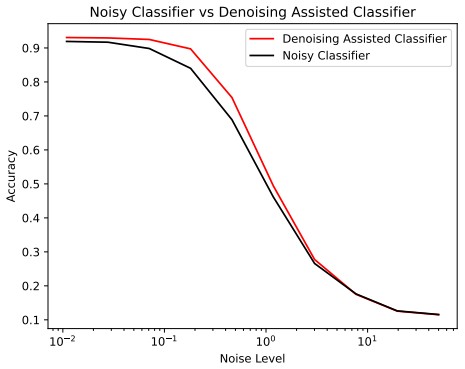

Figure 1: CIFAR10: Test Accuracy vs. Noise Scale.

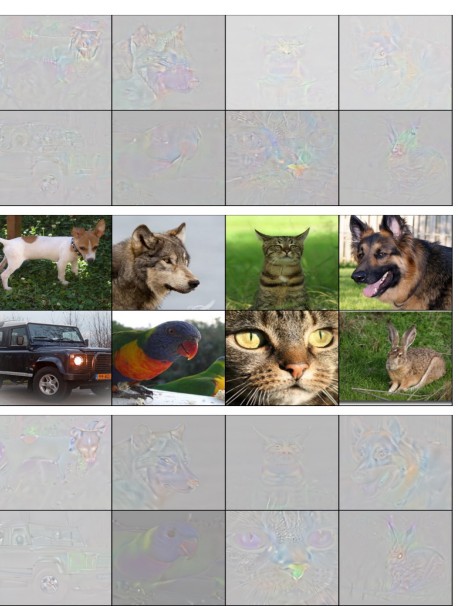

Figure 2: Min-max normalized gradients on samples diffused to $t = 300$ ($T = 999$). Top panel: gradients obtained with noisy classifier. Bottom panel: gradients obtained with DA-classifier. Middle panel: corresponding clean Imagenet samples. We recommend zooming in to see differences between gradients, e.g. the clearer coherence in DA-classifier gradients. This is also shown in Figure 4.

512 using AdamW optimizer with a learning-rate of 3e-4 and weight decay of 0.05. For CIFAR10 classifiers, we use the Exponential Moving Average of the parameters with decay-rate equal to 0.999.

**Classification Accuracy.** We first compare diffusion classifiers in terms of test accuracies as a measure of their generalization. We compute accuracy aggregated over all diffusion times by sampling examples from the forward diffusion seeded with test examples, ensuring that classifiers are evaluated on the same set of diffused examples. We summarize accuracies for CIFAR10 and Imagenet in Table 1 and plot accuracy as a function of noise-scale for CIFAR10 in Figure 1. We find that DA-classifiers generalize better to the test data; this is particularly interesting as random-perturbation training is often used to improve generalization, e.g., Chapelle et al. (2000) discuss Vicinal Risk Minimization for improved generalization by considering Gaussian Vicinities. We hypothesize that denoised images can be viewed as augmentations of the original image that are relatively easier to classify as compared to their noisy counterparts while being diverse enough to improve generalization. Alternatively, introducing the denoising step before classifying the noisy example can be seen as an inductive bias for classifying noisy examples; on its own, cross-entropy loss does not encourage the classifier to *denoise* before assigning a label. To determine the relative importance of noisy and denoised examples in DA-classifiers, we zeroed out one of the input images and measured classification accuracies: while removing either one of the inputs causes the accuracy to drop below the noisy classifier, the drop is higher when the denoised image is masked out.

**Improvements in Generalization** We analyze the improvements in generalization obtained when training on denoised examples instead of noisy examples by comparing these distributions. The denoised example $\hat{\mathbf{x}}$ can be written as

$$\hat{\mathbf{x}} = \mathbb{E}[\bar{\mathbf{x}}_t|\mathbf{x}] = \int_{\bar{\mathbf{x}}_t} \bar{\mathbf{x}}_t \ p_t(\bar{\mathbf{x}}_t|\mathbf{x})d\bar{\mathbf{x}}_t \quad (7)$$

where, $\bar{\mathbf{x}}_t = \mu(\mathbf{x}_0, t)$; see Appendix A for the derivation. In other words, the denoised example $\hat{\mathbf{x}}$ is the expected value of the mean at time $t$, $\bar{\mathbf{x}}_t$, with respect to its conditional probability given the input, i.e. $p_t(\bar{\mathbf{x}}_t|\mathbf{x})$. This is similar to MixUp augmentation (Zhang et al., 2018) wherein

convex combinations of two random examples are used as inputs and the target labels are adjusted accordingly for loss computation; in this case, however, the denoised example is a weighted mean over *perceptually similar* examples and target labels are left unchanged. That is, $p_t(\bar{\mathbf{x}}_t|\mathbf{x})$ in Eq (7) gives more weight to those means $\bar{\mathbf{x}}_t$ that could have been perturbed by gaussian noise to generate $\mathbf{x}$. For smaller noise scales, the distribution $p_t(\bar{\mathbf{x}}_t|\mathbf{x})$ will therefore be more concentrated around perceptually similar examples, and becomes more entropic as the noise scale increases. We can also describe the denoised image $\hat{\mathbf{x}}$ as being *vicinal* to the examples which are likely to have generated $\mathbf{x}$. Furthermore, the number of examples considered for computing the denoised example depends on the noise level and we illustrate this in Figure 8 considering a 2D distribution.

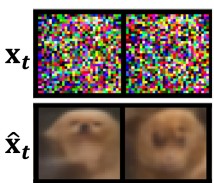

We also illustrate this intuition on high-dimensional data with denoised images generated from an image of a CIFAR10 dog where we observe in each denoised image the superposition of two candidate dog images, i.e. the exact mixup effects described above. To explain the improvements in generalization in both supervised and semi-supervised settings, we interpret training on denoised examples as a type of Vicinal Risk Minimization that is perceptually aligned, and therefore superior to vicinal examples generated by gaussian noise.

**Classifier Gradients (Empirical Observation)**. The diffusion classifiers participate in the sampling through their gradients and we qualitatively compare the classifiers in terms of their class-likelihood gradients $\nabla_{\mathbf{x}} \log p_\phi(y|\mathbf{x}, t)$; for the DA-classifier, we use the total derivative of $\log p_\phi(y|\mathbf{x}, \hat{\mathbf{x}}, t)$ with respect to $\mathbf{x}$. We show min-max normalized gradients from Imagenet and CIFAR10 classifiers in Fig (2) and Fig (3) respectively: in particular, we diffuse clean samples such that some semantic features are still preserved and use these for computing gradients. Intuitively, these gradients should point in the direction that maximizes log-likelihood of the target class. As can be seen in the figures, the gradients obtained from the DA-classifier are more structured and semantically aligned with the clean image than the ones obtained with the noisy classifier. Gradients obtained from classifiers trained only with uncorrupted samples from the dataset are usually unintelligible whereas gradients obtained from classifiers trained with random-smoothing (Kaur et al., 2019) or adversarial-perturbations (Tsipras et al., 2019; Santurkar et al., 2019; Elliott et al., 2021) are perceptually-aligned; Kawar et al. (2023) use this to motivate use of adversarially-robust classifiers for diffusion guidance. We find that introducing a denoising step before classification provides qualitatively superior gradients even without using adversarially perturbed examples for training (see Figures 2, 3, and 4). Qualitatively superior gradients tend to indicate better generalization as the classification system learns to identify an object as a whole. While gradients backpropagated through the denoising module have been used in the past, we observe qualitative benefits of gradients obtained by backpropagating through the denoising module, e.g., Nie et al. (2022) backpropagate through the diffusion SDE for creating adversarial examples and Ho et al. (2022) use derivatives through the denoising module for reconstruction-guided sampling used in autoregressive video diffusion.

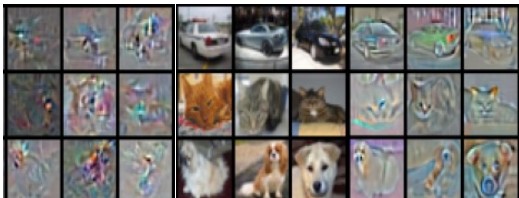

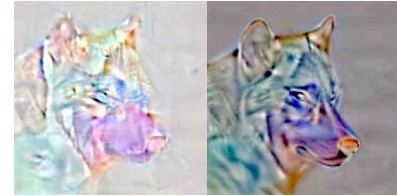

Figure 3: Min-max normalized gradients on samples diffused to $t = 0.35$ ($T = 1.0$). Left panel: gradients obtained with noisy classifier. Right panel: gradients obtained with the DA-classifier. Middle panel: clean corresponding CIFAR10 samples.

Figure 4: To help the reader see the difference between gradients, we show the wolf gradients from Fig (2) in which we applied an identical enhancement to both images, i.e. contrast maximization. The right image shows a DA-classifier gradient.

**Classifier Gradient Analysis.** To understand perceptual alignment of DA-classifier gradients, we compute the total derivative of class-conditional likelihood with respect to the noisy input $\mathbf{x}$:

$$\frac{d\log p_\phi(y|\mathbf{x}, \hat{\mathbf{x}}, t)}{d\mathbf{x}} = \frac{\partial\log p_\phi(y|\mathbf{x}, \hat{\mathbf{x}}, t)}{\partial\mathbf{x}} + \frac{\partial\log p_\phi(y|\mathbf{x}, \hat{\mathbf{x}}, t)}{\partial\hat{\mathbf{x}}} \frac{\partial\hat{\mathbf{x}}}{\partial\mathbf{x}} \tag{8}$$

Empirically, we trace observed improvements in perceptual alignment to the second term (see Fig. 7), and explain these significant improvements as follows:

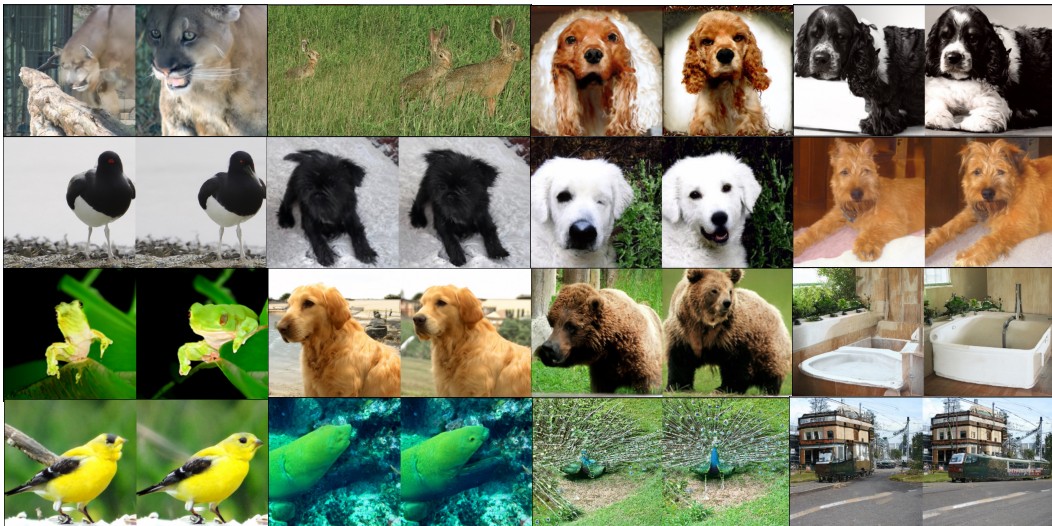

Figure 5: Qualitative Comparison of Diffusion Classifiers on the Image Generation Task using DDIM-100 with same random seed. In each pair, the first image is generated with the Noisy Classifier and the second image is generated with the Denoising-Assisted (DA) Classifier. We observe that the Denoising-Assisted (DA) Classifier improves overall coherence as compared to the Noisy Classifier.

**Theorem 1** *For optimal parameters $\theta$, if $\hat{\mathbf{x}} = \mathbf{x} + \sigma^2(t)s_\theta(\mathbf{x}, t)$, then*

$$\frac{\partial \hat{\mathbf{x}}}{\partial \mathbf{x}} = J = \frac{1}{\sigma^2(t)}\text{Cov}[\bar{\mathbf{x}}_t|\mathbf{x}] \qquad \text{(see proof in Appendix A)}$$

This theorem shows us that the multiplication by $\frac{\partial \hat{\mathbf{x}}}{\partial \mathbf{x}}$ in Eq (8) is in fact a transformation by the covariance matrix $\text{Cov}[\bar{\mathbf{x}}_t|\mathbf{x}]$. This explains the improved perceptual alignment in Fig (3) and Fig (2), since multiplying a vector by $\text{Cov}[\bar{\mathbf{x}}_t|\mathbf{x}]$ *stretches* the vector along the principal directions of the conditional distribution[1] $p(\bar{\mathbf{x}}_t|\mathbf{x})$. We note that our derivation complements Proposition 1 in Chung et al. (2022) which proves certain properties (e.g., $J = J^\top$) of this derivative. In practice, however, the score-function is parameterized by unconstrained, flexible neural architectures that do not have exactly symmetric jacobian matrices $J$. For more details on techniques to enforce conservative properties of score-functions, we refer the reader to Chao et al. (2023).

Table 2: Quantitative comparison of Diffusion Classifiers on the Image Generation Task using 50k samples.

| Dataset Method | CIFAR10 | | | | Imagenet | | | | |
|---|---|---|---|---|---|---|---|---|---|
| | FID↓ | IS↑ | P↑ | R↑ | FID↓ | sFID↓ | IS↑ | P↑ | R↑ |
| Noisy Classifier | 2.81 | 9.59 | 0.64 | 0.62 | 5.44 | **5.32** | 194.48 | 0.81 | 0.49 |
| Denoising Assisted Classifier | **2.34** | **9.88** | 0.65 | 0.63 | **5.24** | 5.37 | **201.72** | 0.81 | 0.49 |

**Image Generation.** We now present an evaluation of diffusion classifiers in the image generation task. We sample 50k images for comparing the diffusion classifiers: in particular, we use a PC sampler as described in Song et al. (2021) with 1000 discretization steps for CIFAR10 samples while we use a DDIM (Song et al., 2020a) sampler with 25 discretization steps for Imagenet samples. We use the 256x256 class-conditional diffusion model open-sourced by Dhariwal and Nichol (2021a) for our Imagenet experiments and set the classifier scale $\lambda_s = 2.5$ following their experimental setup for DDIM-25 samples. The classifier-scale is set to 1.0 for CIFAR10 experiments. Our results (Table 2) show that our proposed Denoising-Assisted (DA) Classifier improves upon the Noisy Classifier in terms of FID and IS for both CIFAR10 and Imagenet at roughly same Precision and Recall levels.

---

[1]To see this, consider the eigen-decomposition $J = U\Lambda U^\top$ where $U$ is an orthogonal matrix of eigenvectors and $\Lambda$ is a diagonal matrix of eigenvalues: if we denote $\mathbf{v} = \frac{\partial \log p_\phi(y|\mathbf{x}, \hat{\mathbf{x}}, t)}{\partial \hat{\mathbf{x}}}$, the product $U\Lambda U^\top \mathbf{v}$ can be analyzed step by step to see that $\mathbf{v}$ gets stretched along the principal directions of the conditional distribution, thus yielding perceptually aligned gradients.

Further, we analyse conditional generation abilities by computing precision, recall, density and coverage for each class, and find that DA-classifiers outperform Noisy classifiers (see Table 4). For example, DA-classifiers yield classwise density and coverage of about 0.92 and 0.77 respectively as compared to 0.78 and 0.71 obtained with Noisy-Classifiers. To qualitatively analyse benefits of the DA-classifier, we generated Imagenet samples using DDIM-100 sampler with identical random seeds and $\lambda_s = 2.5$. In our resulting analysis, we consistently observed that the DA-classifier maintains more coherent foreground and background as compared to the Noisy Classifier. We show examples in Figure 5. We attribute this to the improvements in the classifier gradients presented above. This can also be seen in Figure 4.

## 4 SCORE-SSL: SEMI-SUPERVISED CONDITIONAL SCORE MODELS

In our experiments so far, we have assumed all examples are labelled. However, obtaining a large labeled dataset can be costly, requiring significant manual effort. In this section, we build upon generalization abilities of the DA-classifier to describe a framework for learning Conditional Score Models with partial supervision: specifically, the training data consists of a largely unlabelled dataset, along with a relatively small number of labelled examples.

Semi-supervised learning algorithms for generative models and discriminative models usually differ due to architectural constraints (e.g., invertibility in normalizing flows) and loss-objectives (e.g., adversarial loss for GANs, ELBO loss for VAEs). To achieve our end-goal of semi-supervised conditional generation, however, we require training a diffusion classifier and this allows us to incorporate key ideas from the state-of-the-art semi-supervised classification algorithms: on the other hand, semi-supervised training of a classifier-free model is not straightforward to implement: for example, Ho and Salimans (2022) recommend training with class-label conditioning for 80-90% of each batch and null-token conditioning (i.e., no class label) for the remaining examples whereas we assume that the class-labels are available for less than 10% of the complete training dataset. The general strategy in semi-supervised classification methods is to bootstrap the learning process using labeled data and to utilize unlabeled data along with their label *guesses* as additional labeled training samples; to prevent overfitting, training is often accompanied by consistency losses, regularization, and data augmentations. Consistency regularization is a successful semi-supervised training method which regularizes model outputs on unlabeled examples to be invariant under noise (Xie et al., 2020). While gaussian noise is a typical choice for implementing consistency, state-of-the-art semi-supervised algorithms such as UDA(Xie et al., 2020), FixMatch(Sohn et al., 2020) and MixMatch (Berthelot et al., 2019) use strong augmentations (i.e., heavily distorted images) or MixUp augmentations to generate the noised images. We do not consider these augmentations despite their potential towards increasing classification accuracy because their effect on image generation in the fully-supervised case would need to be carefully analysed first – similar to our analysis of using denoised images as additional inputs in Section 3 – and leave this exploration for future work.

Consider a time-conditional classifier network $p_\phi : \mathbb{R}^{D+1} \to \mathbb{R}^C$ that takes $\mathbf{x} \in \mathbb{R}^D$ and $t \in [0, T]$ as input. We describe the algorithm as it applies for a noisy classifier, and note that extending it to a DA-classifier simply requires adding $\hat{\mathbf{x}}$ in the appropriate spots in the text below, e.g. in Eq (9) and (10). Let $\mathbf{x}_L$ and $\mathbf{x}_U$ denote the sets of labeled and unlabeled samples respectively. For labeled samples, the cross-entropy loss is comprised of two terms, both of which use ground truth labels: (1) one term consists of the loss based on samples diffused to a uniformly sampled time $t \sim \mathcal{U}(0, T)$; (2) the other consists of a loss based on samples diffused to a small time $\tau$ that ensures some semantic features are preserved. The second term mitigates cases when the uniformly sampled time $t$ heavily distorts *most* of the labeled images, which would make it harder for the model to learn from the labeled batch.

$$\mathcal{L}_{\text{CE}}^L = \mathbb{E}_{t,\mathbf{x}}[-\log p_\phi(y|\mathbf{x}, t)] + \mathbb{E}_{\mathbf{x}}[-\log p_\phi(y|\mathbf{x}, \tau)] \tag{9}$$

where, $t \sim \mathcal{U}(0, T)$, $(\mathbf{x}_0, y) \sim \mathbf{x}_L$ and $\mathbf{x} \sim p_t(\mathbf{x}|\mathbf{x}_0)$.

FixMatch recommends using weak-augmentations such as random spatial shifts and random horizontal flips (where applicable) to obtain pseudo-labels for unlabeled examples: in our case, we consider samples diffused to $\tau$ for obtaining pseudo-labels. Following FixMatch, predictions that have confidence greater than some threshold $\eta$ are used as pseudo-labels in computing classification loss on the unlabeled examples diffused to uniformly sampled time $t$. Furthermore, we also consider pseudo-labels from samples diffused to a uniformly sampled time $s$ if the predictions have confidence above $\eta$:

$$\mathcal{L}_{\text{CE}}^U = \mathbb{E}_{t,s,\mathbf{x}}[-\mathbb{1}\{p_\phi(y_\tau|\mathbf{x}_\tau, \tau) \geq \eta\} \log p_\phi(y_\tau|\mathbf{x}, t)] - \mathbb{1}\{p_\phi(y_s|\mathbf{x}_s, s) \geq \eta\} \log p_\phi(y_s|\mathbf{x}, t)] \tag{10}$$

where, $t, s \sim \mathcal{U}(0, T)$, $\mathbf{x}_0 \sim \mathbf{x}_U$, $\mathbf{x} \sim p_t(\mathbf{x}|\mathbf{x}_0)$, $y_\tau = \max_y p_\phi(y|\mathbf{x}_\tau, \tau)$, $y_s = \max_y p_\phi(y|\mathbf{x}_s, s)$ and $\mathbb{1}$ is a binary indicator function that is 1 if the network predicts $y_\tau$ with a confidence greater than $\eta$. Pseudo-labels from confident predictions on samples diffused to random uniform time $s$ could be more informative than samples diffused to time $\tau$ when: (1) additive gaussian noise in samples diffused to time $s$ helps improve uncertainty estimation, and (2) denoised images obtained from unlabeled examples with additive gaussian noise are confidently classified as compared to the original image. At each training step, we estimate pseudo-labels from confident predictions on samples diffused to both $\tau$ and $s$ for computing the cross-entropy loss. We do not resolve disagreements between these two pseudo-labels because we expect that the confident predictions on diffused samples should become consistent over training. In all our experiments; we set $\tau = 0.01$ and set confidence threshold $\eta = 0.95$ following FixMatch.

**Experiments** We evaluate our framework on MNIST, SVHN and CIFAR10 datasets in the classic semi-supervised setting and compare with both generative and discriminative models trained in a semi-supervised setting. We use SSL-VAE and FlowGMM as baselines for generative semi-supervised methods and $\Pi$ Model (Rasmus et al., 2015), Pseudo-Labelling (Lee et al., 2013), Mean Teacher (Lee et al., 2013), MixMatch (Berthelot et al., 2019) and FixMatch (Sohn et al., 2020) as baselines for discriminative semi-supervised methods. We used the VE-SDE for the forward-diffusion as defined in Song et al. (2021) with noise scale $\sigma_t$ ranging from 0.01 to 50.0. We use NCSN++ network for the unconditional score network $s_\theta$: for MNIST and SVHN, we train a 62.8M parameter network for learning $s_\theta$ while we used the pretrained checkpoint of the deeper NCSN++ net-

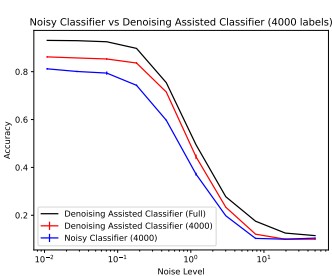

Figure 6: DA Classifier vs Noisy Classifier in Semi-supervised Setting: We find that the Denoising assisted classifier generalizes better than the noisy classifier and in fact, gets very close to the fully-supervised denoising-assisted classifier in some regions.

work containing 107M parameters for CIFAR10 (open-sourced by Song et al. (2021)). For the classifier network $p_\phi$, we use WideResNet 28-2 with 1.5M parameters. We trained the classifier networks using the AdamW optimizer with a learning rate of 3e-4 and weight decay set to 0.05: for MNIST and SVHN, we trained the network for 100k steps while we trained the network for 200k steps for CIFAR10. For all datasets, we used a labeled batch-size of 64 and unlabeled batch-size of 192. We use the DA-classifier described in the Section 3 for all the experiments.

Table 3: Semi-supervised Classification Accuracy: The table shows the semi-supervised classification accuracies with $n_l$ labels. The first block includes semi-supervised generative models as baselines whereas the third block includes accuracies from standard semi-supervised discriminative models for reference. We also include accuracy of a fully-supervised DA-classifier for reference. We use $t = 0.001$ for computing accuracies with diffusion classifiers.

| | Dataset ($n_l/n_u$) | | |
|---|---|---|---|
| Method | MNIST ($1k/59k$) | SVHN ($1k/72k$) | CIFAR10 ($4k/46k$) |
| SSL-VAE (Kingma et al., 2014) | 97.6 | 63.98 | - |
| FlowGMM (Izmailov et al., 2019) | **99.0** | 86.44 | 80.9 |
| Score-SSL (Ours) | **99.1**±0.2 | **96.2**±0.3 | **86.2**±0.2 |
| Denoising-Assisted (DA) Classifier (All labels) | 99.3 | 97.5 | 93.1 |
| $\Pi$ Model (Rasmus et al., 2015) | - | 92.46±0.36 | 85.99±0.38 |
| Pseudo-Labelling (Lee et al., 2013) | - | 90.96±0.61 | 83.91±0.28 |
| Mean Teacher (Tarvainen and Valpola, 2017) | - | 96.58±0.07 | 90.81±0.19 |
| MixMatch (Berthelot et al., 2019) | - | 96.50±0.10 | 93.58±0.10 |
| FixMatch (Sohn et al., 2020) | - | 97.72±0.11 | 95.74±0.05 |

The semi-supervised classification accuracies are summarized in Table 3: we report the average over 3 runs. We observe that our model outperforms the generative modeling baselines in terms of classification accuracy while remaining competitive with discriminative semi-supervised models

(the latter are of course not relevant for direct comparison, but provided as reference points). For comparison, we also trained a noisy classifier on CIFAR10 and plot the accuracy curves in Figure 6: on the clean examples, the DA-classifier obtains an accuracy of about 86%. In our evaluation of the generated images, we find that FID and IS metrics are similar to the results described in Table 2 and note that while the average classwise precision, recall, density and coverage metrics drop for semi-supervised classifiers relative to a fully-supervised classifier, the denoising-assisted classifier outperforms noisy-classifier (see Table 5). We also demonstrate effective results on Positive-Unlabeled settings and include experiments and results in Appendix C.

## 5 RELATED WORK

**Guided Diffusion** Broadly, guidance methods can be categorized into Classifier-free Guidance (CFG) (Ho and Salimans, 2022) and classifier-guidance (CG) (Dhariwal and Nichol, 2021a) depending on whether conditioning information is baked into diffusion model training or done separately by training a classifier. CFG is a popular alternative as it does not require training a separate classifier, but it is inflexible to class re-definitions and requires expensive retraining of the diffusion model from scratch. On the other hand, vanilla classifier-guidance is flexible and potentially modular, but is associated with a range of inadequacies including challenges with generalization and classifier-gradients. Recent efforts towards improved classifier guidance such as Denoising Likelihood Score Matching (DLSM) (Chao et al., 2022) for Score-matching SDEs, and Entropy-Driven (ED) Sampling and Training (Zheng et al., 2022) and Robust-Guidance (Kawar et al., 2023) for DDPMs propose extending the vanilla cross-entropy loss with additional new loss terms. In contrast, Denoising-Assisted (DA) Classifier can be seen as an architectural modification which accepts both noisy and denoised inputs and is *complementary* to DLSM, ED and Robust-Guidance (and indeed exploring these extensions to DA-classifiers constitutes a natural direction for future work) — for reference, we include a comparison of DA-Classifiers with DLSM (Table 6) and ED-Training (Table 7). While our present work is motivated to use denoised examples for improved generalization and qualitatively improved classifier gradients, Universal Guidance (Bansal et al., 2023) explores the use of denoised examples for applying off-the-shelf guidance networks. In a similar vein, Chung et al. (2022) and Chung et al. (2023) explore the use of denoised examples in solving inverse problems: although these works are motivated differently, our empirical and theoretical analyses complement and deepen our understanding of their findings.

**Semi-Supervised Generative Nets** Kingma et al. (2014) represents one of the early works on semi-supervised training of joint classifier and generative models with a Variational Autoencoder. FlowGMM is an elegant method for training Normalizing Flows in a semi-supervised setting wherein they propose to maximize marginal likelihoods for unlabeled data and maximize class-conditional likelihoods for labeled data. D2C (Sinha et al., 2021), Diffusion-AE (Preechakul et al., 2022), and FSDM (Giannone et al., 2022) represent some of the recent efforts on few-shot diffusion models. D2C is a latent-variable conditioned decoder whereas Diffusion-AE is a latent-variable conditioned Diffusion model: in order to introduce class-conditioning, these models train a classifier using the frozen latent-representations and use rejection-sampling for class-conditional generation. Similar to Diffusion-AE, FSDM is a latent-variable conditioned diffusion model that uses Vision Transformer to encode a set of images into a conditioning vector. While FSDM does not support inference of classes on test examples, the classification accuracy in D2C and Diffusion-AE is limited as the latent-variable encoder is frozen and cannot be fine-tuned without retraining the entire pipeline. In contrast, we use a vanilla diffusion model with a flexible classifier architecture to introduce conditioning.

## CONCLUSION

In this work, we propose to improve classifier-guidance with Denoising-Assisted (DA) Diffusion Classifiers wherein both noisy and denoised examples are given as simultaneous inputs. In our experiments, we find that DA-classifiers improve upon noisy classifiers in terms of (a) generalization to test-data, (b) structured and perceptually-aligned classifier gradients, and (c) the image generation task. We also support our empirical observations with theoretical analyses. We reuse the pretrained score-network for obtaining the denoised-inputs used in training the classifier. Finally, we propose and evaluate a framework for semi-supervised training of diffusion classifiers; we find that the generalization abilities of DA-classifiers make them better semi-supervised learners than noisy classifiers. Our evaluations show that the test-accuracies are better than previous semi-supervised generative models and comparable to state-of-the-art semi-supervised discriminative methods.

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

## LIMITATIONS

The score-matching SDE framework allows one to flexibly define a forward diffusion SDE whose score function may not align with the denoising direction. The equivalent implementation of the DA-classifier in these cases could be to provide the predicted score as an additional input to the classifier: however, the improvements introduced by this method would need to be assessed on a case-by-case basis and we do not consider these cases in this work. Sampling from diffusion models requires iterative denoising and is slower as compared to GANs and VAEs; computing classifier-gradients is slower for DA-classifiers as this additionally requires backpropagating through the score-network – recent advances in distilling diffusion models for fast sampling (Luhman and Luhman, 2021) such as Consistency-Models (Song et al., 2023) and Progressive Distillation (Salimans and Ho, 2022) could be extended for class-conditional diffusion models.

## A   DERIVATION OF THEOREM 1

For the forward-diffusion SDEs considered in this paper, as discussed in Section 2, the marginal distribution $p_t(\mathbf{x})$ can be expressed in terms of the data distribution $p(\mathbf{x}_0)$:

$$p_t(\mathbf{x}) = \int_{\mathbf{x}_0} p_t(\mathbf{x}|\mathbf{x}_0)p(\mathbf{x}_0)d\mathbf{x}_0 \qquad (11)$$

where

$$p_t(\mathbf{x}|\mathbf{x}_0) = \mathcal{N}(\mathbf{x} \mid \mu(\mathbf{x}_0, t), \ \sigma^2(t)\mathbf{I}).$$

If we denote $\mu(\mathbf{x}_0, t)$ by $\bar{\mathbf{x}}_t$, so that $p(\bar{\mathbf{x}}_t)$ represents the distribution over the mean at time $t$, we can rewrite $p_t(\mathbf{x})$ as

$$p_t(\mathbf{x}) = \int_{\bar{\mathbf{x}}_t} p_t(\mathbf{x}|\bar{\mathbf{x}}_t) p(\bar{\mathbf{x}}_t) d\bar{\mathbf{x}}_t$$

since $\mu$ is linear and invertible. The optimal score-function $s_{\theta*}(\mathbf{x}, t) = \nabla_{\mathbf{x}} \log p_t(\mathbf{x})$ can be simplified as:

$$s_{\theta*}(\mathbf{x}, t) = \frac{1}{p_t(\mathbf{x})} \int_{\bar{\mathbf{x}}_t} \frac{\bar{\mathbf{x}}_t - \mathbf{x}}{\sigma^2(t)} p_t(\mathbf{x}|\bar{\mathbf{x}}_t) p(\bar{\mathbf{x}}_t) d\bar{\mathbf{x}}_t \tag{12}$$

Using Eq (12), we can rewrite the denoised example, $\hat{\mathbf{x}} = \mathbf{x} + \sigma^2(t)s_\theta(\mathbf{x}, t)$, as:

$$\hat{\mathbf{x}} = \frac{1}{p_t(\mathbf{x})} \int_{\bar{\mathbf{x}}_t} \bar{\mathbf{x}}_t \, p_t(\mathbf{x}|\bar{\mathbf{x}}_t) \, p(\bar{\mathbf{x}}_t) d\bar{\mathbf{x}}_t = \int_{\bar{\mathbf{x}}_t} \bar{\mathbf{x}}_t \, p_t(\bar{\mathbf{x}}_t|\mathbf{x}) d\bar{\mathbf{x}}_t = \mathbb{E}[\bar{\mathbf{x}}_t|\mathbf{x}] \tag{13}$$

That is, the denoised example $\hat{\mathbf{x}}$ is in fact the expected value of the mean $\bar{\mathbf{x}}_t$ given input $\mathbf{x}$. (See also Eq (7) in the main text ).

To compute $\frac{\partial \hat{\mathbf{x}}}{\partial \mathbf{x}}$, we algebraically simplify $\int_{\bar{\mathbf{x}}_t} \bar{\mathbf{x}}_t \nabla_{\mathbf{x}} \left( \frac{p_t(\mathbf{x}|\bar{\mathbf{x}}_t)}{p_t(\mathbf{x})} \right) p(\bar{\mathbf{x}}_t) d\bar{\mathbf{x}}_t$ as follows:

$$
\begin{aligned}
\frac{\partial \hat{\mathbf{x}}}{\partial \mathbf{x}} &= \int_{\bar{\mathbf{x}}_t} \bar{\mathbf{x}}_t \left( \frac{\nabla_{\mathbf{x}} p_t(\mathbf{x}|\bar{\mathbf{x}}_t)}{p_t(\mathbf{x})} - \frac{p_t(\mathbf{x}|\bar{\mathbf{x}}_t) \nabla_{\mathbf{x}} p_t(\mathbf{x})}{p_t^2(\mathbf{x})} \right)^\top p(\bar{\mathbf{x}}_t) d\bar{\mathbf{x}}_t \\
&= \int_{\bar{\mathbf{x}}_t} \bar{\mathbf{x}}_t \left( \frac{p_t(\mathbf{x}|\bar{\mathbf{x}}_t)}{p_t(\mathbf{x})} \frac{\bar{\mathbf{x}}_t - \mathbf{x}}{\sigma^2(t)} - \frac{p_t(\mathbf{x}|\bar{\mathbf{x}}_t) \nabla_{\mathbf{x}} \log p_t(\mathbf{x})}{p_t(\mathbf{x})} \right)^\top p(\bar{\mathbf{x}}_t) d\bar{\mathbf{x}}_t \\
&= \int_{\bar{\mathbf{x}}_t} \bar{\mathbf{x}}_t \frac{p_t(\mathbf{x}|\bar{\mathbf{x}}_t)}{p_t(\mathbf{x})} \left( \frac{\bar{\mathbf{x}}_t - \mathbf{x}}{\sigma^2(t)} - \nabla_{\mathbf{x}} \log p_t(\mathbf{x}) \right)^\top p(\bar{\mathbf{x}}_t) d\bar{\mathbf{x}}_t \\
&= \int_{\bar{\mathbf{x}}_t} \bar{\mathbf{x}}_t \frac{p_t(\mathbf{x}|\bar{\mathbf{x}}_t)}{p_t(\mathbf{x})} \left( \frac{\bar{\mathbf{x}}_t - \mathbf{x} - \sigma^2(t) \nabla_{\mathbf{x}} \log p_t(\mathbf{x})}{\sigma^2(t)} \right)^\top p(\bar{\mathbf{x}}_t) d\bar{\mathbf{x}}_t \\
&= \int_{\bar{\mathbf{x}}_t} \bar{\mathbf{x}}_t \frac{p_t(\mathbf{x}|\bar{\mathbf{x}}_t)}{p_t(\mathbf{x})} \left( \frac{\bar{\mathbf{x}}_t}{\sigma^2(t)} - \frac{\mathbf{x} + \sigma^2(t) \nabla_{\mathbf{x}} \log p_t(\mathbf{x})}{\sigma^2(t)} \right)^\top p(\bar{\mathbf{x}}_t) d\bar{\mathbf{x}}_t \\
&= \int_{\bar{\mathbf{x}}_t} \frac{\bar{\mathbf{x}}_t \bar{\mathbf{x}}_t^\top}{\sigma^2(t)} p_t(\bar{\mathbf{x}}_t|\mathbf{x}) d\bar{\mathbf{x}}_t - \left( \int_{\bar{\mathbf{x}}_t} \bar{\mathbf{x}}_t \, p_t(\bar{\mathbf{x}}_t|\mathbf{x}) d\bar{\mathbf{x}}_t \right) \frac{\hat{\mathbf{x}}^\top}{\sigma^2(t)} \\
&= \frac{1}{\sigma^2(t)} \left( \int_{\bar{\mathbf{x}}_t} \bar{\mathbf{x}}_t \bar{\mathbf{x}}_t^\top p_t(\bar{\mathbf{x}}_t|\mathbf{x}) d\bar{\mathbf{x}}_t - \hat{\mathbf{x}} \hat{\mathbf{x}}^\top \right) \\
&= \frac{1}{\sigma^2(t)} \left( \mathbb{E}[\bar{\mathbf{x}}_t \bar{\mathbf{x}}_t^\top|\mathbf{x}] - \mathbb{E}[\bar{\mathbf{x}}_t|\mathbf{x}] \mathbb{E}[\bar{\mathbf{x}}_t|\mathbf{x}]^\top \right) \\
&= \frac{1}{\sigma^2(t)} \mathrm{Cov}[\bar{\mathbf{x}}_t|\mathbf{x}]
\end{aligned}
\tag{14}
$$

## B    CLASSWISE EVALUATION OF CONDITIONAL GENERATION METRICS

Table 4: Evaluation of Fully Supervised CIFAR10 Diffusion Classifiers: Average Classwise Precision, Recall, Density and Coverage Metrics computed with 5k generated images per class.

| Method | P ↑ | R ↑ | D ↑ | C ↑ |
|---|---|---|---|---|
| Noisy Classifier | 0.57 | 0.62 | 0.78 | 0.71 |
| Denoising-Assisted Classifier | **0.63** | **0.64** | **0.92** | **0.77** |

$$\frac{d \log p_\phi}{d\mathbf{x}} \quad \frac{\partial \log p_\phi}{\partial \mathbf{x}} \quad \frac{\partial \log p_\phi}{\partial \hat{\mathbf{x}}} \quad \frac{\partial \log p_\phi}{\partial \hat{\mathbf{x}}} \frac{\partial \hat{\mathbf{x}}}{\partial \mathbf{x}}$$

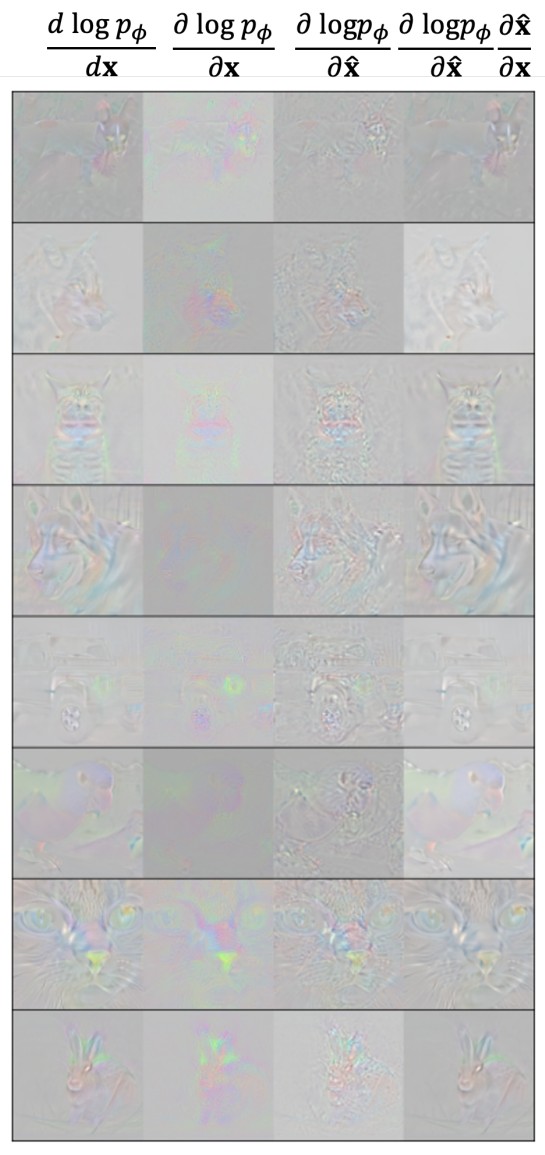

Figure 7: The figure shows the total derivative $\frac{d \log p_\phi}{d\mathbf{x}}$ (Eq. 8), the partial derivative with respect to noisy input $\frac{\partial \log p_\phi}{\partial \mathbf{x}}$, the partial derivative with respect to denoised input $\frac{\partial \log p_\phi}{\partial \hat{\mathbf{x}}}$, and $\frac{\partial \log p_\phi}{\partial \hat{\mathbf{x}}} \frac{\partial \hat{\mathbf{x}}}{\partial \mathbf{x}}$.

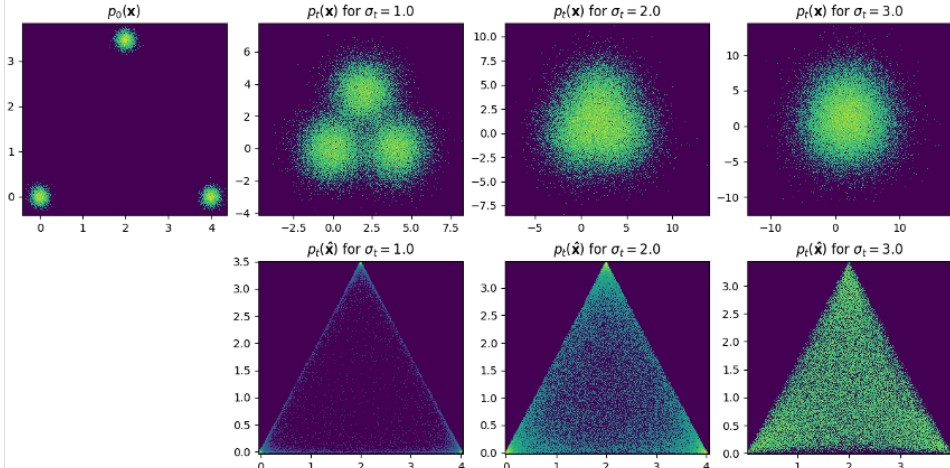

Figure 8: Illustration of Noisy Distributions and Denoised Distributions over 2D Synthetic Data: Here we consider $p_0(\mathbf{x})$ to be a distribution of 3 Gaussians placed at the vertices of an equilateral triangle. Considering a VE-SDE whose $\sigma_{\max}$ is set to the side-length of the triangle (following Song and Ermon (2020)), we compare $p_t(\mathbf{x})$ and $p_t(\hat{\mathbf{x}})$ for different choices of $\sigma_t$ as shown: we generate $\hat{\mathbf{x}}$ by empirically estimating $p_t(\mathbf{x})$ and computing the score-function as $\nabla_{\mathbf{x}} \log p_t(\mathbf{x})$. We note that the denoised examples are a convex sum over the means $\bar{\mathbf{x}}_t$ that are likely to have generated $\mathbf{x}$. At $\sigma_t = 1.0$, we see that almost none of the denoised points consider a convex sum over examples from all 3 clusters and with larger noise scales, more examples are assigned a non-zero weight in the convex sum.

Table 5: Evaluation of Semi-Supervised CIFAR10 Diffusion Classifiers trained with 4000 labeled examples: Average Classwise Precision, Recall, Density and Coverage Metrics computed with 5k generated images per class. Both classifiers were shown the same labeled portion.

| Method | FID ↓ | IS ↑ | P ↑ | R ↑ | D ↑ | C ↑ |
|---|---|---|---|---|---|---|
| Noisy Classifier | 2.82 | 9.61 | 0.44 | 0.62 | 0.52 | 0.45 |
| Denoising-Assisted Classifier | **2.35** | **9.86** | **0.55** | **0.64** | **0.64** | **0.57** |

Table 6: DLSM vs DA-Classifier: In this table, we directly compare between using DLSM – i.e., DLSM-Loss in addition to cross-entropy loss in training classifiers on noisy images as input – and DA-Classifiers wherein we use both noisy and denoised images as input but only used cross-entropy loss for training. We compare between FID, IS and average classwise Precision, Recall, Density and Coverage metrics. While the FID and IS scores are comparable, we note that our class-wise Precision, Recall, Density and Coverage metrics are either comparable or demonstrate a significant improvement.

| Method | FID ↓ | IS ↑ | P ↑ | R ↑ | D ↑ | C ↑ |
|---|---|---|---|---|---|---|
| DLSM | **2.25** | **9.90** | 0.56 | 0.61 | 0.76 | 0.71 |
| Denoising-Assisted Classifier | 2.33 | 9.88 | **0.63** | **0.64** | **0.92** | **0.77** |

Table 7: ED vs DA-Classifier: Zheng et al. (2022) propose two complementary techniques to improve over vanilla classifier-guidance: Entropy-Constraint Training (ECT) and Entropy-Driven Sampling (EDS). ECT consists of adding an additional loss term to the cross-entropy loss encouraging the predictions to be closer to uniform distribution (similar to the label-smoothing loss). EDS modifies the sampling to use a diffusion-time dependent scaling factor designed to address premature vanishing guidance-gradients. The sampling method (EDS/Vanilla) can be chosen independent of the training method (determined by the loss-objective and classifier-inputs). In the following, we compare between ECT and DA-Classifiers using Vanilla Sampling method using the results in Table 3 of Zheng et al. (2022). We observe that DA-Classifier obtains better FID/IS than ECT.

| Method | Loss-Objective | Classifier-Inputs | Sampling Method | FID | sFID | IS |
|--------|----------------|-------------------|-----------------|-----|------|-----|
| Noisy-Classifier | CE | Noisy Image | Vanilla | 5.46 | 5.32 | 194.48 |
| ECT-Classifier | CE+ECT | Noisy Image | Vanilla | 5.34 | **5.3** | 196.8 |
| DA-Classifier | CE | Noisy Image & Denoised Image | Vanilla | **5.24** | 5.37 | **201.72** |

## C   POSITIVE-UNLABELED EXPERIMENTS

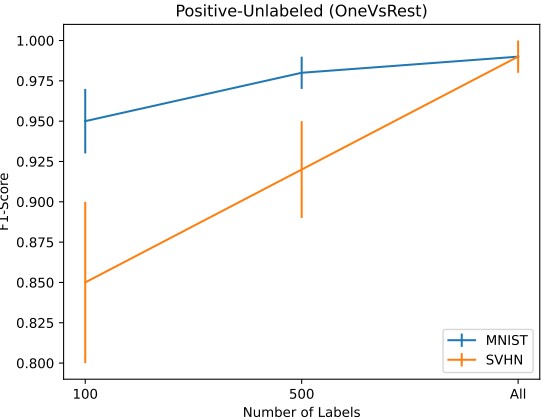

Figure 9: Positive-Unlabeled Learning Results: The graph shows the F1-scores on MNIST and SVHN for OneVsRest Positive-Unlabeled training setup. Specifically, we select one of the 10 classes as positive, label 100 or 500 of them and treat the remaining as negative. We report the mean and variance of the F1-score across 10 models. For reference, we also show the F1-score when the entire training data is available.

Positive-unlabeled setting involves learning a binary classifier trained on labeled positive samples, together with an unlabeled training set containing samples from both positive and negative classes. Typically, a positive-unlabeled learning problem is solved in two steps: a) estimating proportions of positive and negative classes (the mixture proportion estimation step); and b) training a binary classifier using this information. In the following experiments, we follow previous work (e.g., Acharya et al. (2022)) and primarily focus on training a binary classifier assuming that the mixture proportion estimation step has already been performed and the class prior is known.

Table 8: Classification accuracy results on PU-MNIST: We randomly choose 1k examples of Odd digits as positive examples and treat the rest as unlabeled. We repeat the experiment 3 times.

| | PU-MNIST (OddvsEven) |
|--|--|
| PvU (Elkan and Noto, 2008) | 91.10±0.92 |
| uPU (Niu et al., 2016) | 91.14±0.87 |
| nnPU (Kiryo et al., 2017) | 91.83±0.79 |
| puNCE (Acharya et al., 2022) | 94.7±0.19 |
| Score-SSL(Ours) | **98.8±0.05** |

In the positive unlabeled setting, we additionally minimize the cross-entropy between the class-averages obtained on unlabeled examples diffused to time $\tau$ and the supplied class prior. If $q(y)$ denotes the supplied class prior distribution of class $y$ and $r(y) = \mathbb{E}_{\mathbf{x}_U}[p_\phi(y|x,\tau)]$, then we additionally minimize the cross-entropy between $q$ and $r$ computed as $-\sum_y q(y) \log r(y)$.

**Experiments.** We conduct Positive-Unlabeled experiments on MNIST and SVHN by selecting one of the 10 classes as the positive class. We report the F1-scores for different proportions of labels in Figure 9: we observe that our model generalizes well given few positive examples and class prior. We also compare our model accuracy on MNIST with other PU baselines in Table 8: here, the classifier is trained to classify between odd and even digits and 1k odd examples are provided as positive samples.

# D    UNCURATED SAMPLES

In the following, we include:

- CIFAR10 samples generated with diffusion classifiers trained on fully labeled dataset.
- CIFAR10, MNIST and SVHN samples generated with denoising-assisted classifier trained on a partially labeled dataset along with a larger unlabeled dataset. We also show generated samples for different settings of classifier-scale.

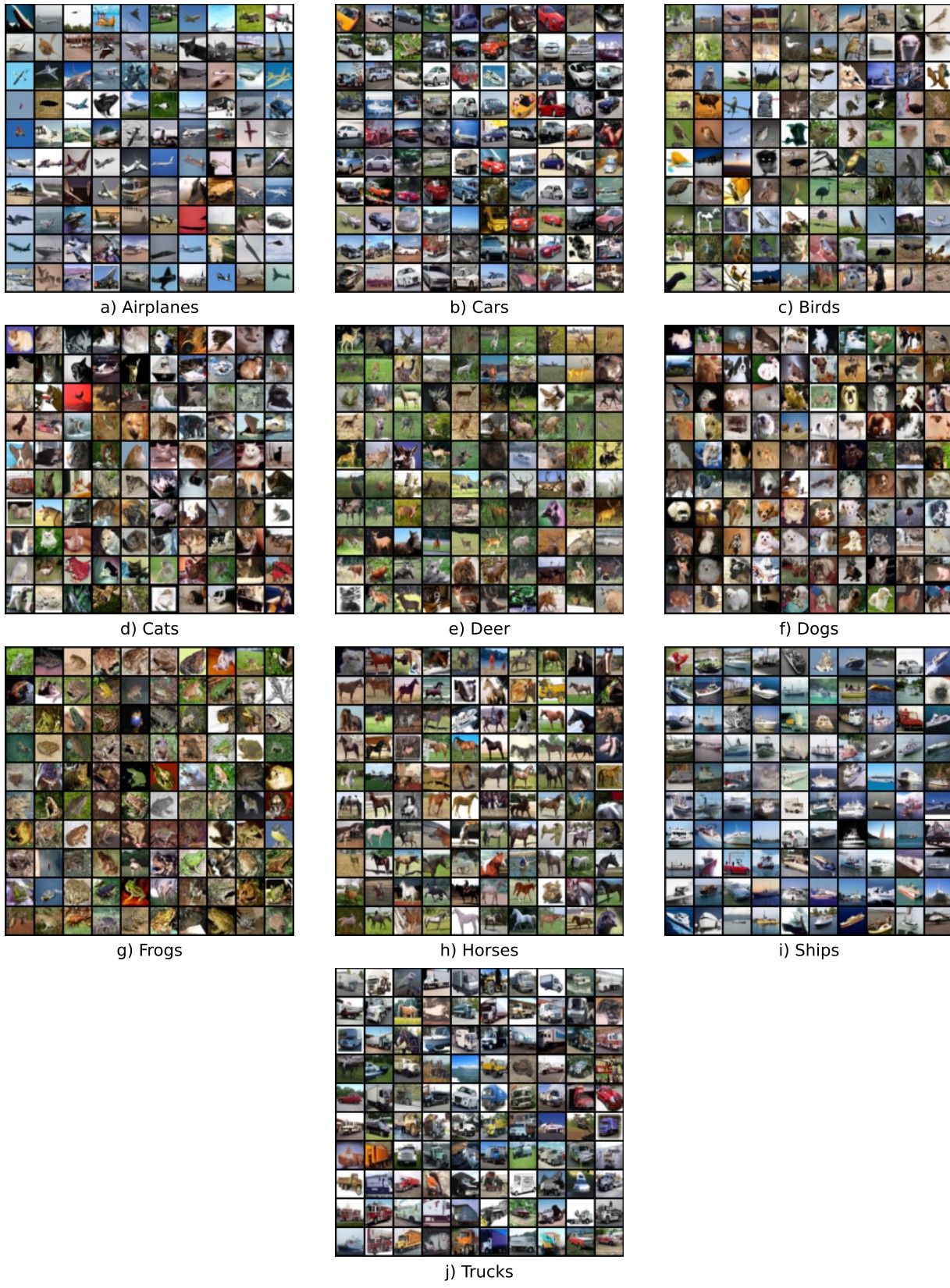

Figure 10: Uncurated CIFAR10 Samples with Noisy-Classifier (Full training data).

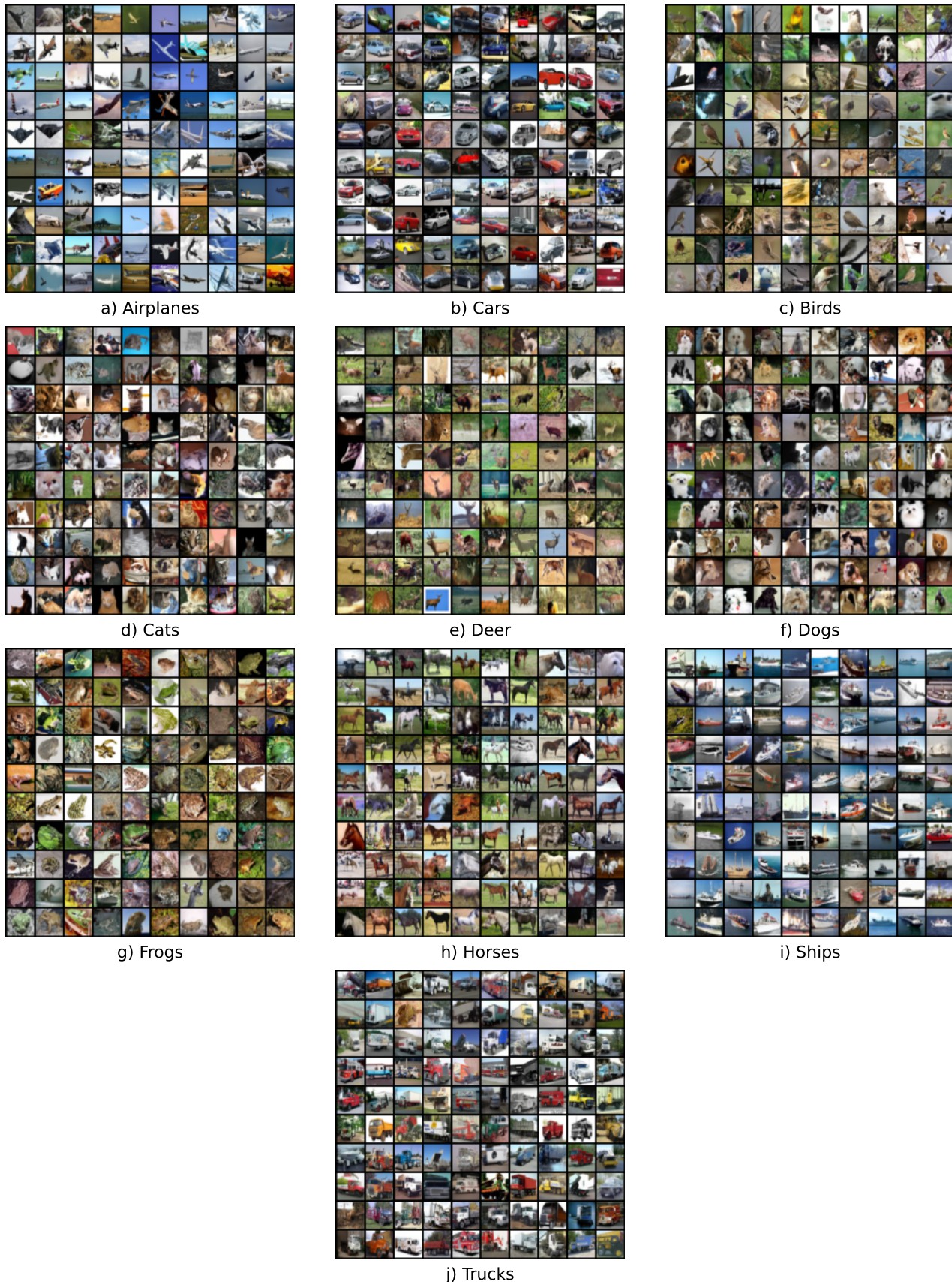

Figure 11: Uncurated CIFAR10 Samples with DA-classifier (Full training data).

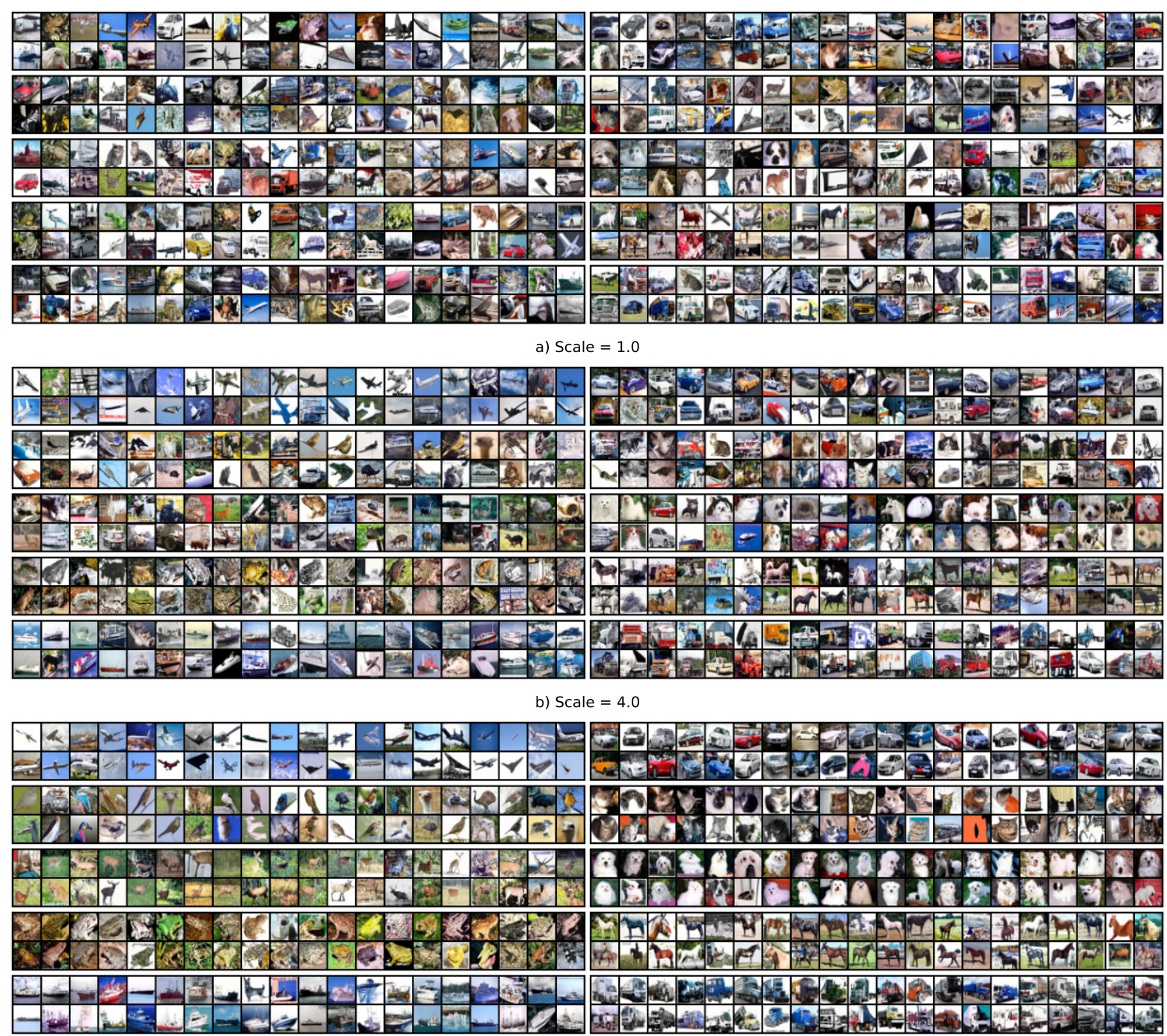

a) Scale = 1.0

b) Scale = 4.0

c) Scale = 10.0

Figure 12: Uncurated samples generated with Denoising-Assisted CIFAR10 classifier trained with 4k labels for different settings of classifier scale.

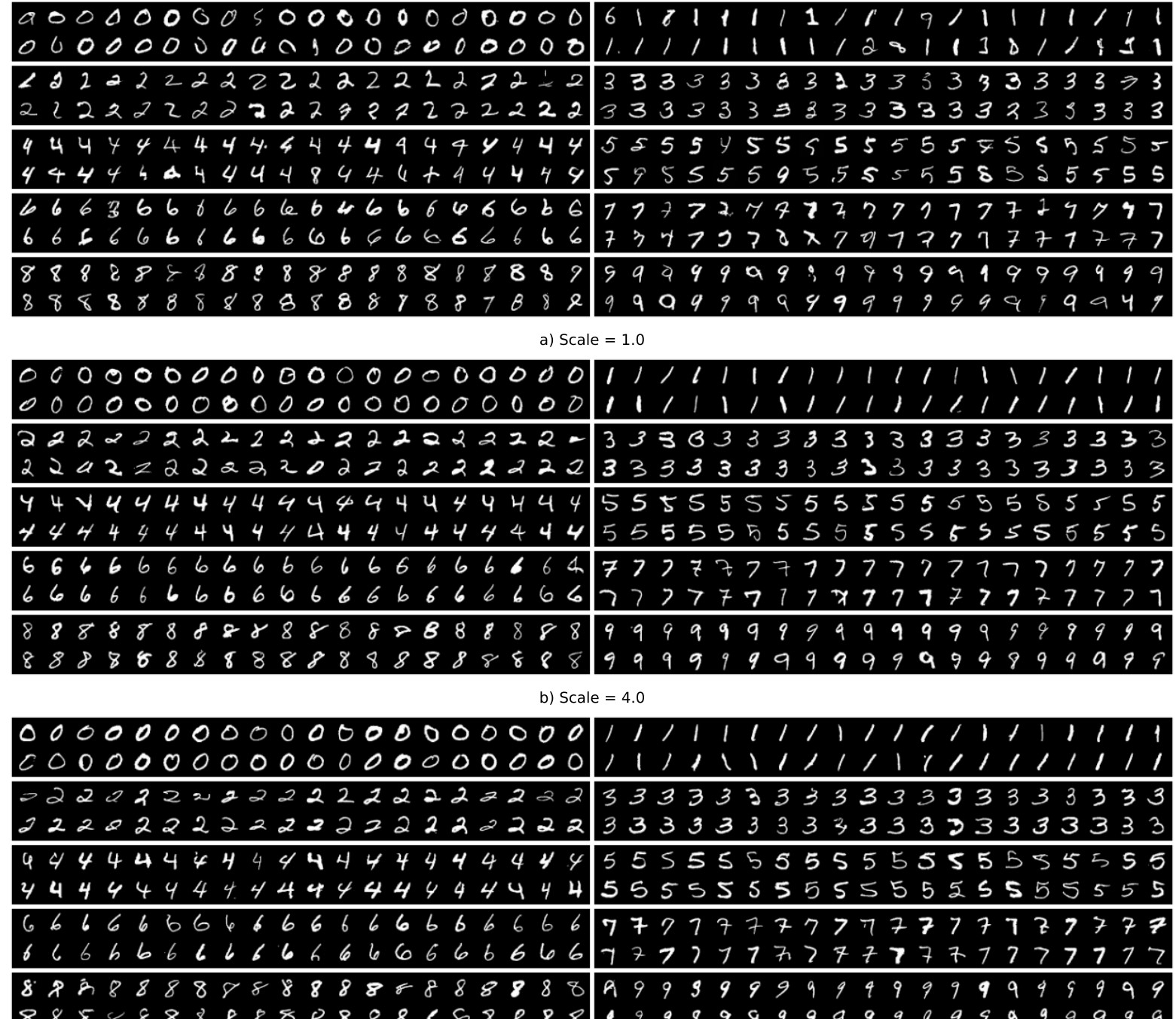

a) Scale = 1.0

b) Scale = 4.0

c) Scale = 10.0

Figure 13: Uncurated samples generated with Denoising-Assisted MNIST classifier trained with 1k labels for different settings of classifier scale.

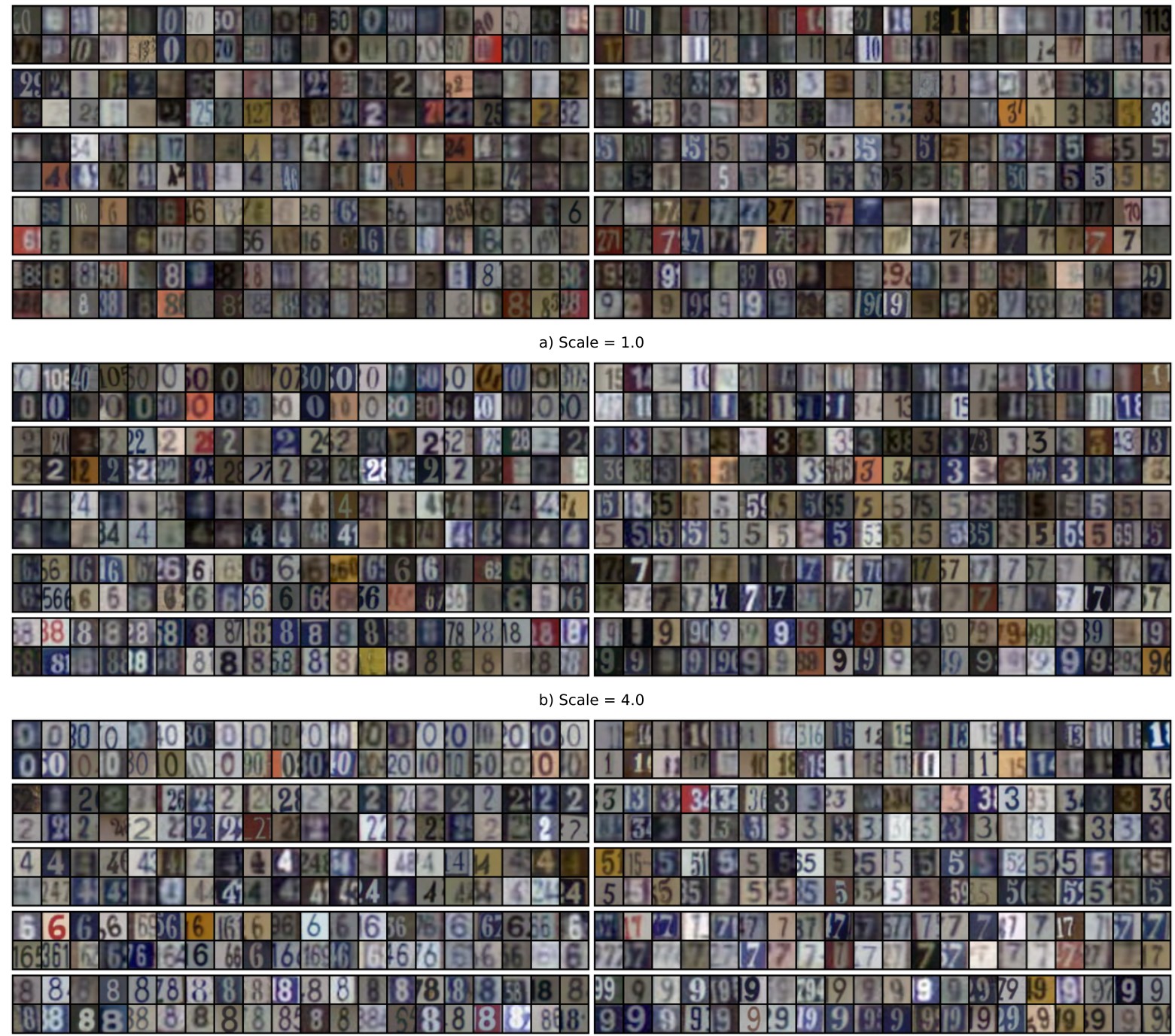

a) Scale = 1.0

b) Scale = 4.0

c) Scale = 10.0

Figure 14: Uncurated samples generated with Denoising-Assisted SVHN classifier trained with 1k labels for different settings of classifier scale.

