# OpenReview forum: "Training Diffusion Classifiers with Denoising Assistance"
_ICLR.cc/2024/Conference — Submitted to ICLR 2024_

### Official Review · Reviewer_JmSU · 2023-10-13

**Soundness:** 2 fair
**Presentation:** 2 fair
**Contribution:** 2 fair
**Rating:** 5
**Confidence:** 4

**Summary:**

The paper introduces Denoising-Assisted (DA) classifiers in the domain of classifier-guided diffusion models to enhance both conditional and unconditional generation tasks. The DA classifiers are trained using both noisy and denoised examples, unlike traditional diffusion classifiers trained only on noisy data. Through experiments on CIFAR10 and Imagenet datasets, the authors demonstrate that DA classifiers exhibit better generalization to unseen data and improved perceptual alignment of classifier-gradients, leading to enhanced image generation. The paper also theoretically analyzes the gradients of DA-classifiers to explain the observed improvements. Additionally, a semi-supervised framework is proposed to leverage the generalization strengths of DA-classifiers in scenarios with limited labeled data. The empirical and analytical discussions included in the paper provide a thorough understanding of the improvements DA-classifiers bring over noisy classifiers, showing promise in advancing the performance of generative models.

**Strengths:**

The paper unfolds an innovative notion of utilizing denoised samples as inputs to the classifier within a diffusion model framework. The simplicity of this idea is elegantly juxtaposed with its effectiveness, as substantiated by the authors through empirical evaluations and theoretical elucidations. The inner workings of the proposed method are also thoughtfully explained, shedding light on the mechanisms that contribute to its efficacy.

**Weaknesses:**

1. The presentation of the paper could be improved for better readability. Specifically, the formatting of equations (5) and (6) spanning across three rows appears to be cluttered and may benefit from a more concise representation.

2. The paper primarily focuses on one-step denoising, which raises the question of why multi-step denoising to reach time step t=0 was not explored. Utilizing the sample at time step t=0 as input to the classifier could potentially offer additional insights or improvements, and it would be beneficial for the authors to discuss or explore this aspect.

3. The core contribution of employing the denoised sample as input to a classifier may come across as straightforward. The paper could benefit from a clearer articulation of the motivation behind this choice and the specific problems it aims to address. While the one-step denoised sample is utilized, the rationale behind not exploring [x, one-step denoised sample, two-step denoised sample] as inputs could use further clarification. Although the authors provide some explanations, a more robust justification could enhance the perceived significance of the work.

4. The discussion on the semi-supervised learning framework introduces an idea of selecting pseudo-label data based on confidence thresholds during the diffusion process. However, the motivation behind this choice could be better elucidated. It may be worth exploring or explaining why traditional uncertainty measures in the raw data space were deemed insufficient, and how the proposed method addresses any identified limitations.

**Questions:**

See Weaknesses.

---

> ### Author Response · Authors · 2023-11-17
> **Response to Reviewer JmSU**
>
> We thank the reviewer for their insightful and helpful comments for improving the paper.
>
> 1) Presentation: Thank you for the feedback. We will fix the formatting in Eqs. (5,6), and we gladly welcome any other suggestions.
>
> 2) Motivation for using single-step denoised examples. We use denoised example $\hat{\bf x_t} = {\bf x_t}+\sigma^{2}(t) s_\theta({\bf x_t},t)$
> as an additional input to the classifier network because (a) learning to classify denoised image is potentially easier than learning to classify noisy examples and (b) we have access to a pretrained score network $s_\theta({\bf x_t},t)$ that can help us estimate the additive gaussian noise contained in ${\bf x}_t$. Beyond just simplifying the learning task, we observe that this simple change also helps address the following deficiencies of vanilla classifier guidance: 1) Generalisation in fully-supervised and semi-supervised settings, 2) Perceptual-alignment of Classifier-gradients, and 3) Image Generation Performance. We support these findings both theoretically and empirically. Additionally, we also show significant advantages of using denoised-images in semi-supervised training as opposed to simple gaussian-noise augmented images. We will clarify this motivation in the revised paper.
>
> 3) Multi-step denoising. As described in Eq. (4), the unconditional score function estimated by $s_\theta({\bf x_t},t)$ is adjusted with the gradient of log-likelihood predicted by the classifier in classifier-guided diffusion sampling, as $s_\Theta({\bf x_t}, t|y) = \nabla_{{\bf x_t}} \log p_\phi(y|{\bf x_t},t) + s_\theta({\bf x_t},t)$. Here, $\theta$ refers to parameters of the unconditional score function, $\phi$ refers to parameters of the classifier function and $\Theta = \\{\theta, \phi\\}$. The suggestion of using the sample at $t-2$ or $t=0$ for estimating the score-function $s_\Theta({\bf x_t}, t|y)$ is indeed very interesting but requires separate theoretical and empirical investigation; in particular, it may require making appropriate modifications to the sampling algorithm. In this study, we consider using denoised example obtained at time $t$ as an additional input to the classifier for estimating $s_\Theta({\bf x_t}, t|y)$ and this does not require any modification to the sampling algorithm.
>
> 3) Uncertainty measures for pseudo-label estimation. To train the diffusion classifier on partially-labelled data, we need to estimate the pseudo-labels of the unlabeled examples in each batch. As described in the paper, we follow FixMatch and use confidence-thresholding, a traditional measure of uncertainty, to estimate the pseudo-labels. In order to estimate the pseudo-labels, we use two types of data samples from forward diffusion to estimate the pseudo-labels — samples diffused to 1) $\tau=0.01$ (i.e., with negligible gaussian noise) and 2) uniformly sampled time $s$. Since the samples diffused to time $\tau=0.01$ have negligible gaussian noise, that is similar to estimating pseudo-labels from raw data using traditional uncertainty measures. In addition, we also use pseudo-labels from confident predictions on samples diffused to random uniform time $s$ as they could be more informative than raw data in the following cases:
>    * Additive gaussian noise in samples diffused to time $s$ could potentially help the noisy-classifier/DA-classifier better estimate the uncertainty in labels as compared to samples containing negligible gaussian noise.
>    * Denoised images obtained from unlabeled examples with additive gaussian noise could potentially help classification, even when the DA-classifier is not able to confidently classify the original raw image.
>
>    At each training step, we estimate pseudo-labels from confident predictions on samples diffused to both $\tau$ and $s$ and use them for computing the cross-entropy loss. We do not resolve disagreements between these two pseudo-labels because we expect that the confident predictions on diffused samples should become consistent over training. We thank the reviewer for this comment and we will clarify this discussion in the revised text.
>
> We look forward to addressing any outstanding concerns in the discussion period.

---

> > ### Comment · Reviewer_JmSU · 2023-11-21
> > **Thanks for the feedback.**
> >
> > Thanks for the authors feedback. It clarified some concerns. Yet, I did not get the contribution of the "4 SEMI-SUPERVISED CONDITIONAL SCORE MODELS". What is the novelty here?

---

> > > ### Author Response · Authors · 2023-11-23
> > > **Response to Reviewer JmSU**
> > >
> > > Thank you very much for taking the time to read our rebuttal and respond to it.
> > >
> > > The key novelty in section 4 is rooted in our demonstration of improvements in semi-supervised training using denoised examples; to the best of our knowledge, we are not aware of any other work that focuses on semi-supervised classifier-guidance or uses denoised examples for semi-supervised training. To demonstrate this, we extended FixMatch for semi-supervised training of diffusion classifiers. In the following, we restate the goal of Section 4, provide a summary of our contributions towards realizing this goal, and then articulate our main contributions in more detail:
> > >
> > > Goal: Given a partially labeled dataset, train a classifier to classify samples from the forward-diffusion.
> > >
> > > Summary: In fully supervised settings, a batch of images from the dataset is first diffused to a randomly sampled time $t$ and the classifier is trained to predict the label of the image that was used to obtain the diffused example. Semi-supervised training typically consists of training on labelled examples along with a consistency regularization loss on unlabeled examples that is motivated to ensure identical predictions on perturbed versions derived from the same example. We extend FixMatch to develop a semi-supervised training method that closely resembles the fully-supervised training of diffusion classifiers except that we use pseudo-labels for computing training losses on unlabeled examples. We show that using denoised examples as additional inputs improves generalization in semi-supervised settings resulting in reliable pseudo-labels (as compared to noisy examples alone) which in turn lead to improved test accuracies.
> > >
> > > Our contributions are as follows:
> > >
> > > 1) *We extend FixMatch for semi-supervised training of diffusion classifiers:* Of the several strategies available for imposing consistency loss, we choose to build on top of FixMatch due to its simplicity: FixMatch uses confident predictions on weakly augmented examples to obtain pseudo-labels and uses these pseudo-labels to train on strongly-augmented (i.e., heavily distorted images) unlabeled examples – in other words, FixMatch constrains the model to be invariant to strong augmentations. In turn, we consider confident predictions on examples containing negligible gaussian noise for computing training losses on unlabeled examples diffused to uniformly sampled time $t$. In addition, we also consider confident predictions on examples diffused to a (different) uniformly sampled time $s$ for the reasons described above – this can also be interpreted as imposing consistency in predictions on two samples from the forward diffusion.
> > >
> > >       * Note regarding strongly-augmented examples: As discussed in the paper, we do not consider these augmentations despite their potential towards increasing classification accuracy because their effect on image generation in the fully-supervised case would need to be carefully analysed first – similar to our analysis of using denoised images as additional inputs in Section 3 – and leave this exploration for future work. In other words, since strong-augmentations significantly alter the original data distribution, it is important to first analyse the role of these augmentations on image generation for fully supervised settings.
> > >
> > > 2) *We show that improved generalisation observed in fully-supervised settings through using denoised examples as additional inputs transfers to semi-supervised settings.* Generalisation is important to ensure successful semi-supervised training of a classifier: for example, if a classifier confidently predicts incorrect classes, then including these examples in training will amplify its (incorrect) biases leading to poorer test-accuracies. In Fig. (6), we demonstrate improvements of the DA-Classifier over the Noisy Classifier in terms of the test-accuracy as a function of diffusion time. This provides a key novel contribution of section 4 and is particularly interesting because random perturbations with gaussian noise have been traditionally used for consistency regularizations in semi-supervised training and we show that using denoised examples instead can lead to better generalization for diffusion classifiers. We also explain these generalization effects by drawing connections to MixUp augmentation and vicinal risk minimization in the paragraph “Improvements in Generalization” on Page 4.
> > >
> > > Thank you for the opportunity to highlight this novel contribution, and we will ensure that it is sufficiently emphasized in the paper.

---

### Official Review · Reviewer_F98v · 2023-10-26

**Soundness:** 2 fair
**Presentation:** 2 fair
**Contribution:** 2 fair
**Rating:** 5
**Confidence:** 2

**Summary:**

This manuscript proposed the denoising-assisted (DA) classifier that employs additional denoised image for better classification. The DA classifier has shown better performance on CIFAR-10 and ImageNet compared to original diffusion classifiers, and this imporvement is also verified by theoretical analysis.

**Strengths:**

1. A new method of DA classifier is proposed by employing the denoised image for better classification;

2. Emprical experiments on CIFAR-10 and ImageNet verifies the effectiveness of the proposed approach;

3. Some insightful observations in terms of the classifier gradients are presented to analyse the imporvement of DA classifier.

**Weaknesses:**

IMHO, the writing and organization of this manuscirpt can be greatly imporved for clearer illustration. For example, the motivation and the most related works w.r.t. diffusion classifiers are quilte unclear. Moreover, the writing are redundant and blunt, which blends the authors's contributions and existed works. Some technical questions are presented as below:

1. What is the motivation to develop DA classifier?  What is the advantages of DA classifier compared to existed classifiers including conventional deep models (VGG, ResNet, etc.) or large models (CLIP, etc.)?

2. What the authors do is to employ the denoised images as the additional input. To this end, the authors should provide more detail about how to get the denoised images and analysing the influence of different types of denoised images.

3. Why Theorem 1 explain the improvements of DA classifier?

Minors and typos:
1. "Cifar10" -> "CIFAR-10"
2. "Imagenet" -> "ImageNet"
3. Page 3, Sec. 2.2: $y\in [1,C]$ -> $y\in =\\{1, ..., C\\}$

4. Page 3, Sec. 3: "propose to use as input both ..." -> "propose to use both ... as input"
5. To many long sentences.

**Questions:**

See Weaknesses.

---

> ### Author Response · Authors · 2023-11-17
> **Response to Reviewer F98v**
>
> We thank the reviewer for their comments. We will incorporate the minor suggestions & typo corrections in our revision. We address the remaining points as follows:
>
> > What is the motivation to develop DA classifier?
>
> We systematically expose several deficiencies  of vanilla classifier guidance,
> 1) Generalization in fully-supervised and semi-supervised settings,
> 2) Perceptual-alignment of Classifier-gradients,
> 3) Image Generation Performance
>
> and we demonstrate how simply using denoised images as additional input can address all these deficiencies. We call these denoising-assisted (DA) classifiers.
>
>
> > What is the advantages of DA classifier compared to existed classifiers including conventional deep models (VGG, ResNet, etc.) or large models (CLIP, etc.)?
>
> We believe that this question is asking for some overall context, so here is an overview: For classifier-guided diffusion, we train classifiers on images sampled from the forward diffusion, which generally contain additive gaussian noise. As described in the paper, a DA-Classifier refers to a classifier that takes both noisy and denoised images as simultaneous inputs, in contrast to a noisy-classifier that only receives noisy images as inputs. Noisy Classifier and DA-Classifier can be constructed with any backbone neural architecture: as described in the paper, we use WideResNet-28-2 for all CIFAR10, MNIST and SVHN experiments (following Song et al., 2021) while we use the downsampling-half of the UNET for ImageNet experiments (following Dhariwal and Nichol (2021b) ). The vanilla diffusion classifier and the corresponding denoising-assisted diffusion classifier have the same network architecture after the first layer and differ only in terms of the first convolution module: for example, if the first input-convolution of vanilla diffusion classifier takes an input volume of $(N, C_i, H_i, W_i)$ and gives an output volume of $(N, C_o, H_o, W_o)$, the DA-Classifier takes an input volume of $(N, 2C_i, H_i, W_i)$ and gives the same size output $(N, C_o, H_o, W_o)$.
>
> > “the authors should provide more detail about how to get the denoised images and analysing the influence of different types of denoised images”
>
> As described below Eq. 6, we obtain denoised images $\hat{\bf x}$ using the pretrained score-network $s_\theta$ as $\hat{\bf x} = {\bf x}+\sigma^{2}(t) s_\theta({\bf x},t)$. We would be glad to provide additional details about this in an appendix, but we do not understand exactly what additional details would be helpful for this reader. We are also not sure what the reviewer means by “different types of denoised images”; any clarification on this would also be welcome. As explained above, we are denoising images that have been corrupted with additive gaussian noise.
>
> > “Why Theorem 1 explain the improvements of DA classifier?”
>
> As written in Section 3, Theorem-1 “explains the improved perceptual alignment in Fig (3) and Fig (2), since multiplying a vector by $\text{Cov}[{\bf \bar{\bf x}}_t|{\bf x}]$  *stretches* the vector along the principal directions of the conditional distribution $p({\bf \bar{\bf x}}_t|{\bf x})$.” Intuitively, since the conditional distribution $p({\bf \bar{\bf x}}_t|{\bf x})$ corresponds to the distribution of candidate denoised images, the principal directions of variation are perceptually aligned and hence stretching the gradient along these directions will yield perceptually aligned gradients. We attribute the quantitative and qualitative improvements on the image generation task to 1) perceptually aligned classifier gradients and 2) improved generalisation to unseen examples.
>
> We look forward to addressing any outstanding concerns in the discussion period.

---

> > ### Comment · Reviewer_F98v · 2023-11-23
> >
> > Many thanks for the authors' response.

---

### Official Review · Reviewer_YWxN · 2023-10-30

**Soundness:** 3 good
**Presentation:** 3 good
**Contribution:** 3 good
**Rating:** 6
**Confidence:** 3

**Summary:**

This work proposes an approach utilizing both noisy and denoised examples in the diffusion process as inputs for training a classifier (DA-classifier). Comparative analysis against the noisy classifier method demonstrates the effectiveness of the proposed approach. Additionally, by analyzing its generalization, gradients, and image generation quality, the study further explains the efficacy of the proposed approach.

**Strengths:**

1. The paper conducts both quantitative (generalization to test data) and qualitative (perceptually-aligned classifier gradients and generative modeling metrics) analyses, exhibiting the advantages of DA-classifiers.
2. The study not only empirically examines the performance of DA-classifiers but also provides theoretical explanations for their gradient properties, theoretically supporting improved perceptual alignment.
3. The proposed method, though quite simple, proves to be effective in empirical validation.

**Weaknesses:**

1. The numerical results are not good. In Table 2, there are no significant differences between the two methods, especially P and R on the two datasets and all metrics on ImageNet. The visualization in Figures 2, 3, and 4 is impressive.
2. The motivation behind using diffusion-based samples to train a classifier for semi-supervised learning is unclear and requires further clarification. It is okay to see that semi-supervised generative models perform much worse than those discriminative models. Can we see the comparisons in image generations under the semi-supervised settings? Table 5 only shows the results trained on labeled data.

**Questions:**

1. Why the results of Tables 2 and 4 are not consistent.
2. 'Cifar10' in abstract.

---

> ### Author Response · Authors · 2023-11-17
> **Response to Reviewer YWxN**
>
> We thank the reviewer for their insightful comments and summary. We address the mentioned weaknesses and answer the questions as follows:
>
> 1) Numerical results and consistency between metrics in Table 2 and Table 4: Thank you for asking about this. The Precision, Recall, Density and Coverage metrics compare between the manifold of generated distribution and manifold of the source distribution in terms of nearest-neighbours and can be computed conditionally (i.e., classwise) or unconditionally [1]. Table 2 shows the Precision and Recall metrics computed unconditionally while Table 4 shows the Precision, Recall, Density and Coverage metrics computed classwise. For CIFAR10, we do indeed observe significant improvements in terms of classwise Precision, Recall, Density and Coverage metrics (see Table 4). We also observe that using DA-Classifier instead of vanilla classifier improves FID/IS on both ImageNet and CIFAR-10 at similar precision and recall values – this suggests that both the classifiers have similar manifolds in the nearest-neighbours sense but the generated images with DA-Classifier are more realistic than vanilla-classifiers. In Fig. 5, we qualitatively observe that DA-Classifiers improve over Vanilla-classifiers in terms of overall coherence and since these improvements are generally subtle, we do observe slight but meaningful improvements of FID/IS over the baseline. We will incorporate this discussion into the revised text. Overall, we believe that the numbers are indeed consistent with strong performance, but if there is any remaining result or comparison that the reviewer sees as not being good, then if we failed to provide a clear explanation or interpretation of it, we would be grateful to have the opportunity to correct that. (Thank you!)
>
>    Also, thank you for appreciating the strength of our visualizations! While visualizations can be hard to quantify, we are very excited by these results and agree that they demonstrate a significant contribution of this work!
>
> 2) Semi-supervised training: To clarify: the semi-supervised training is exactly analogous to the fully-supervised training in that the classifier is trained to classify samples from the forward diffusion (as described in section 2.2 and Eq. 5); the key difference between the two is that we use pseudo-labels for training over unlabeled examples. Table 5 summarises the image-generation performance of the semi-supervised Noisy-classifier and semi-supervised DA-Classifier trained on the same labeled and unlabeled portions of CIFAR-10: we observe that the semi-supervised DA-Classifier significantly outperforms semi-supervised vanilla-classifier in terms of the classwise Precision, Recall, Density and Coverage metrics. The FID/IS scores in the semi-supervised setting are similar to the fully-supervised case.
>
> We look forward to addressing any outstanding concerns in the discussion period.
>
> [1] Naeem, M.F., Oh, S.J., Uh, Y., Choi, Y. and Yoo, J. Reliable fidelity and diversity metrics for generative models. ICML, 2020.

---

### Official Review · Reviewer_dXPD · 2023-11-01

**Soundness:** 2 fair
**Presentation:** 2 fair
**Contribution:** 2 fair
**Rating:** 3
**Confidence:** 4

**Summary:**

The paper introduces Denoising-Assisted (DA) classifiers to improve the classifier guidance method in diffusion models. DA classifiers are time-dependent classifiers where the input includes denoised examples in addition to perturbed examples and timesteps. The effectiveness of DA classifiers is demonstrated through test classification results, gradient analysis, and the quality of the generated images. They also consider semi-supervised settings, and DA classifiers outperform noisy classifiers in terms of classification accuracy.

**Strengths:**

The proposed method is simple but effective. Incorporating denoised examples into the classifiers is a reasonable approach to improve performance. Additionally, it is worth noting that they address partial supervision settings, although they did not provide the result on image generation performance.

**Weaknesses:**

1. Baseline

The paper lacks sufficient baseline comparisons. To strengthen their argument, the authors should compare their method with other relevant classifier guidance methods such as DLSM, ED, and Robust-Guidance. While they mention that these methods are complementary, it would be beneficial to present their performance results, similar to how standard semi-supervised discriminative models are presented in Table 3. In addition, the authors should extend the application of DA classifiers to these methods and report the results to demonstrate the orthogonal nature of their work.

2. Missing generation results in semi-supervised settings

There is no FID and IS results for generated images in semi-supervised settings. They only said that "FID and IS metrics are similar to the results described in Table 2". Since the main task of this paper could be seen as conditional image generation, these experimental results are crucial evidence to assess the effectiveness of their method.

3. Classifier-free guidance in semi-supervised training

The authors mentioned that semi-supervised training of a classifier-free model is not easy to implement, citing the original paper's recommendation for the supervised settings. It is important to empirically verify this claim, as there is a lack of evidence on for the feasibility and effectiveness of classifier-free models in partial label settings.

4. Clarification of the sampling procedure

Since the primary objective of this paper is image generation, it would be beneficial to provide a detailed procedure or algorithm for sample generation using the proposed method. Actually, the utilization of DA classifiers in sample generation is not straightforward, given the difference in gradient computation between $\nabla_x \log p(y|x,t)$ and $\frac{d \log p(y|x,\hat{x},t)}{dx}$. A discussion of the implications of this difference would help clarify the approach.

**Questions:**

1. Please check the issues in the Weaknesses section.

2. Ablation study

It would be beneficial to include an ablation study to investigate the key factors contributing to the improvement of the DA classifier. Two specific experiments might be considered:
* Changing $\hat{x}$ to $s_\theta (x,t)$ in the DA classifier inputs: this would help to identify the essential elements, such as denoised examples or information of data scores.
* Use only denoised examples, i.e. $p_\phi (y|\hat{x}, t): this would evaluate the importance of using both perturbed and denoised examples.

3. Minor points

* Eq. (7) should explicitly state that this equation holds for the optimal score network.
* The figure in the middle of page 5 lacks a caption.
* In the paragraph "Classifier Gradients (Empirical Observation)", "... classifiers trained only with uncorrupted samples..." may be corrected to "... classifiers trained only with corrupted samples...".
* "Score-SSL" does not appear in the manuscript, except in Table 3.

---

> ### Author Response · Authors · 2023-11-17
> **Response to Reviewer dXPD (Part 1)**
>
> We thank you for your detailed reviews and insightful questions. We will incorporate the minor corrections and suggestions in our revision along with elements of our response below. We address your concerns as follows:
>
> 1) Baselines
>
> Even though DLSM-loss and DA-Classifier represent orthogonal improvements over vanilla classifier, as recognized by the reviewer, we directly compare between the two methods on CIFAR10 (trained on VE-SDE) based on the reviewer’s request:
> ||||||||
> |---|:---:|:---:|:---:|:---:|:---:|:---:|
> |  | FID  | IS  | CW Precision  | CW Recall  | CW Density  | CW Coverage  |
> | **DLSM** | 2.25 | 9.90 | 0.56 | 0.61 | 0.76 | 0.71 |
> | **Ours** | 2.33 | 9.88 | 0.63 | 0.64 | 0.92 | 0.77 |
>
> While the FID and IS scores are comparable (especially in relation to the typical FID/IS scores observed in the literature), we note that our class-wise Precision, Recall, Density and Coverage metrics are either comparable or demonstrate a significant improvement. Although we are very interested to analyse the fusion of each methodology, this is out of scope of this current paper: furthermore, a careful analysis of fusion in terms of generalisation, classifier-gradients and image-generation performance also requires specific experiments for finding optimal hyperparameters for each fusion -- for example, the hyper parameters controlling the adversarial attack in Robust-Guidance, the softmax temperature in Entropy-driven Training/Sampling, and so on. We are excited to present these kinds of studies in a thorough and systematic future work. But in the present paper, our focus is, rather, to comprehensively analyse the benefits of using denoised examples as an additional input to the classifier both empirically and theoretically while only using the cross-entropy loss objective. The suggested baseline methods such as DLSM, Robust-Guidance and Entropy-Driven Training/Sampling propose novel loss-objectives additional to the usual cross-entropy loss and are hence orthogonal to our method in which we propose using denoised images as additional inputs to the classifier; furthermore, while the baselines are focused on improving classifier-guidance in fully-supervised settings, we also consider semi-supervised settings. In our initial exploration, we tested to see if the DLSM loss objective can help in semi-supervised settings but did not observe any improvements over just using the cross-entropy loss. We agree that these comparisons strengthen the paper and we will add these results in our revision.
>
> 2) Semi-supervised generation results
>
> We thank the reviewer for mentioning this and giving us the opportunity to provide additional numerical evaluations. As mentioned in the paper, the FID and IS scores of images generated with semi-supervised classifiers are very similar to the FID/IS scores of images generated with fully-supervised classifiers: specifically, the FID/IS scores of semi-supervised vanilla classifier and semi-supervised DA-classifier are 2.82/9.61 and 2.35/9.86 respectively. We will include these in the revised version of the paper. The class-wise P/R/D/C values are, however, different for semi-supervised classifiers as compared to fully-supervised classifiers and we do include these in Table 5 (in Appendix B) – where, we can see that the DA-Classifier outperforms noisy-classifier in semi-supervised settings as well.

---

> ### Author Response · Authors · 2023-11-17
> **Response to Reviewer dXPD (Part 2)**
>
> 3) Provide empirical evidence that classifier-free guidance in a semi-supervised setting is hard:
>
> We provide a two-part rebuttal to this comment: first, we argue that comparing between classifier-guidance and classifier-free guidance in fully-supervised settings is not straightforward to begin with. Then, we elaborate upon the challenges in implementation of classifier-free guidance in semi-supervised settings that further complicates comparison between the two methods in semi-supervised settings.
>
> (a) classifier-free guidance and classifier-guidance represent two main categories of guidance methods for diffusion models; classifier-guidance is modular, theoretically motivated and allows for flexible conditioning without requiring to retrain the unconditional score model while classifier-free guidance is a heuristic method that bakes the conditioning into the score-model and requires careful hyperparameter tuning [1]. Comparing between classifier-guidance and classifier-free guidance properly is challenging: for example, previous studies such as GLIDE [2] do this very well, but they rely upon extensive (and expensive) human evaluation of generated samples to show that classifier-guidance often qualitatively underperforms classifier-free guidance; this result is in contrast to the quantitative metrics they computed such as CLIP-scores that usually favoured classifier-guided samples over classifier-free samples. We therefore argue that establishing fair comparison criteria between the two methods would require dedicated research efforts that would be beyond the scope of this paper. Here, we focus on addressing the following deficiencies of vanilla classifier guidance by using denoised images as additional input: 1) Generalization in fully-supervised and semi-supervised settings, 2) Perceptual-alignment of Classifier-gradients, 3) Image Generation Performance.
>
> (b) Training a classifier-free diffusion model with partially labelled dataset is harder because the training relies heavily on labels, as described in our text: “…Ho and Salimans (2022) recommend training with class-label conditioning for 80-90% of each batch and null-token conditioning (i.e., no class label) for the remaining examples whereas we assume that the class-labels are available for less than 10% of the complete training dataset.” If we directly follow the suggestion in that paper, it is easy to see that we will overfit to the labeled examples as we recycle examples from the labeled pool more often than the examples in the unlabeled pool:
>
> >In the example of CIFAR10, we assume that 4K examples are labelled out of a total 50k samples. If we consider a batch-size of 512 and select the percentage of labeled examples in a batch to be 80%, we will need 410 labeled examples and 112 unlabeled examples for each gradient step. Overall, this implies that 1 epoch over the unlabeled examples corresponds to 41 epochs over the labeled examples (since, 10 gradient steps correspond to 1 epoch over the labeled data).
>
> Thus, avoiding overfitting to the labelled pool would require separate research efforts–again outside the scope of this paper–-to identify reliable pseudo-labelling strategies that enable a feasible way to follow the recommendation in Ho and Salismans (2022) — in order to get good generative performance.
>
> [1] Bahjat Kawar, Roy Ganz, Michael Elad. Enhancing Diffusion-Based Image Synthesis with Robust Classifier Guidance. TMLR, 2023.
>
> [2] Nichol, A., Dhariwal, P., Ramesh, A., Shyam, P., Mishkin, P., McGrew, B., Sutskever, I. and Chen, M. GLIDE: Towards Photorealistic Image Generation and Editing with Text-Guided Diffusion Models. ICML, 2022.
>
> 4) Sampling procedure
>
> Thank you for raising this point: this gives us an opportunity to highlight the fact that the implementation of sampling procedure with DA-Classifiers is indeed as straightforward as with vanilla-classifiers: the gradient of the log-probability estimated by the classifier is used as additional guidance in both cases. In our implementation, we simply concatenate the noisy and denoised example and use this as input to the classifier. In terms of the model architecture, we simply double the ```in_channels``` of the input convolution for DA-Classifiers and the remaining architecture is left unchanged: in fact, for the Imagenet experiments in the paper, we expanded the in-channels of the first convolution layer and continued training the pretrained model for 50k steps. We will incorporate this into the paper.

---

> ### Author Response · Authors · 2023-11-17
> **Response to Reviewer dXPD (Part 3)**
>
> 5) Ablations
>
> Thank you for suggesting these ablations. As described in the paper, we evaluated the relative importance of noisy and denoised examples in DA-classifiers by zeroing out one of the input images and measuring its effect upon classification accuracies; we found that while removing either one of the inputs causes the accuracy to drop below the noisy classifier, the drop is higher when the denoised image is masked out. This demonstrates that the DA-classifier makes use of both noisy and denoised examples in the classification. In fact, the intention behind using both noisy and denoised images as input is to allow the model to use parts of both images as needed. Upon acceptance, we will also include an ablation study considering the recommended ablations in the appendix.
>
> We look forward to addressing any outstanding concerns in the discussion period.
>
> [1] Bahjat Kawar, Roy Ganz, Michael Elad. Enhancing Diffusion-Based Image Synthesis with Robust Classifier Guidance. TMLR, 2023.
> [2] Nichol, A., Dhariwal, P., Ramesh, A., Shyam, P., Mishkin, P., McGrew, B., Sutskever, I. and Chen, M. GLIDE: Towards Photorealistic Image Generation and Editing with Text-Guided Diffusion Models. ICML, 2022.

---

> ### Comment · Reviewer_dXPD · 2023-11-20
>
> Thank you for your response. However, I still have the following concerns.
>
> 1. Baselines
>
> The authors only compare to the reported values of DLSM among the mentioned baselines. For example, for Table 1 in ED, it appears that they compare the ImageNet experiment with the same settings in the paper, and it seems to perform better than the DA classifier. A comparison and analysis with ED is also necessary.
>
> | | FID | sFID | IS | P | R |
> |:---:|:---:|:---:|:---:|:---:|:---:|
> |Noisy Classifier|5.44|5.32|194.48|0.81|0.49|
> |DA Classifier|5.24|5.37|201.72|0.81|0.49|
> |ED|4.67|5.12|235.24|0.82|0.47|
>
> Additionally, the noisy classifier performance as reported in the manuscript also seems to be the same as the performance reported in the ED paper, so it seems to be extracted from that paper. If the values are reported in the previous work, it would be nice to clearly indicate that.
>
> Also, the authors claim that the proposed method is orthogonal to existing methods, which needs to be supported. This is because it is possible, for example, that the existing methods simply reflect the effect that can be achieved with the Denoising Assistance in a different way. To avoid this concern, the fusion experiments mentioned by the authors are not out of scope, but would be very important to show the orthogonality of the proposed method.
>
> 2. Semi-supervised generation results
>
> As far as I know, a revised version can be posted during the discussion period, so I would like to recommend that the discussion mentioned in the author response be added to the revised version.
>
> 3. Classifier-free guidance
>
> The author claims that the classifier-free diffusion model could easily overfit in partially labeled datasets. As far as I know, there is no experimental evidence for this, and it would be very important to show this experimentally to support the author's opinion. As the author states, it is beyond the scope of this study to find and compare suitable techniques that do not overfit classifier-free guidance. However, I think it is necessary to show experimentally that the current intuitive application of classifier-free guidance leads to overfitting.
>
> 4. Sampling procedure
>
> I understand that the practical implementation can be made not to differ much from the existing classifier guidelines, as the authors mentioned. However, the part I pointed out is that in the DA classifier, there is a noisy sample $x$ and a denoised sample $\hat{x}$, and the denoised sample $\hat{x}$ is computed from $x$. Therefore, I think the derivative for the DA classifier is expected to require a total derivative as I expressed above. I asked for a discussion of how the effect of a change in $x$ is reflected in the denoised sample, and if not, why it shouldn't be.

---

> > ### Author Response · Authors · 2023-11-21
> > **Response to Reviewer dXPD (Part 1)**
> >
> > Thank you very much for responding to our rebuttal!
> >
> > 1) Baselines
> >
> > Zheng et al [1] propose two techniques to improve over vanilla classifier-guidance: Entropy-Constraint Training (ECT) and Entropy-Driven Sampling (EDS). ECT consists of adding an additional loss term to the cross-entropy loss encouraging the predictions to be closer to uniform distribution (similar to the label-smoothing loss). EDS modifies the sampling to use a diffusion-time dependent scaling factor designed to address premature vanishing guidance-gradients. The sampling method (EDS/Vanilla) can be chosen independent of the training method (determined by the loss-objective and classifier-inputs). In order to properly compare between training-methods, we should use identical sampling methods. From results in Table 3 of [1] (included below for reference), we can compare between ECT-classifier and DA-classifier using the same sampling method: in the following, we observe that DA-Classifier obtains better FID/IS than ECT when using the vanilla sampling method.
> >
> > ||||||||
> > |---|:---:|:---:|:---:|:---:|:---:|:---:|
> > | Method | Loss-Objective | Classifier-Inputs | Sampling Method | FID | sFID | IS | P | R |
> > | Noisy-Classifier | CE | Noisy Image | Vanilla | 5.46 | 5.32 | 194.48 | 0.81 | 0.49 |
> > | ECT-Classifier | CE+ECT | Noisy Image | Vanilla | 5.34 | **5.3** | 196.8 | 0.81 | 0.49 |
> > | DA-Classifier | CE | Noisy Image & Denoised Image | Vanilla | **5.24** | 5.37 | **201.72** | 0.81 | 0.49 |
> > |||||
> > | Noisy-Classifier | CE | Noisy Image | EDS | 4.82 | 5.04 | 218.97 | 0.8 | 0.5 |
> > | ECT-Classifier+EDS | CE+ECT | Noisy Image | EDS | 4.67 | 5.12 | 235.24 | 0.82 | 0.48 |
> >
> > We also emphasize that we trained our DA-Classifier on imagenet by fine-tuning the classifier checkpoint — released by Dhariwal and Nichol (2021b) — for 50k steps (batch-size = 128) with denoised-image as an additional input; in contrast, ECT is trained from scratch for 500K steps with a batch size of 256. This further highlights the effectiveness of our proposed method. We thank you for raising this question and allowing us to demonstrate these improvements over ECT. Upon acceptance, we will include results of the DA-Classifier obtained with EDS.
> >
> > **Orthogonality.** We agree that experiments can help empirically determine if two methods are complementary and concretely demonstrate the benefits of combining the two methods. In this case, however, we can  also (deductively) *explain* the complementarity of additional loss-objectives in relation to  denoised examples as additional input. We will first present this reasoning, and then we will address the compute-benefit trade-off as it relates to this specific case.
> >
> > I. Complementarity of two approaches: Consider a DA-Classifier $f: ({\bf x}, \hat{\bf x}) \mapsto {y}$ and a noisy-classifier $g: {\bf x} \mapsto {y}$.
> >
> > 1. Observation 1: If $F$ represents the set of all possible parameterizations of DA-Classifier $f$ and $G$ represents the set of all possible parameterizations of Noisy-classifier $g$, then $G \subsetneq F$ (i.e., $G$ is a proper subset of $F$).
> >
> >    Explanation: All possible functions realised by $G$ can be realised by $F$ by simply setting the weights of the input-convolution corresponding to the denoised input to zero.
> > 2. Observation 2: For a given loss-objective $L$, if $L_F$ and $L_G$ correspond to the optimal validation losses obtained using functions from sets $F$ and $G$ respectively, then $L_F \le L_G$.
> >
> >      Explanation: In the worst case, the network can choose to completely ignore the denoised input if it is easier to minimise $L$ by only considering the noisy input — i.e., the network can flexibly choose to behave similarly to the noisy classifier.
> >
> > From the above two observations, we can see that DA-Classifier can match the Noisy-classifier in the worst case and can potentially improve over the performance obtained by the noisy-classifier for any given loss-function. Exactly how the denoised examples are going to be useful would need to be assessed separately for each loss objective and as already explained above, this is out of scope. We consider cross-entropy loss and explain the advantages of using denoised-examples as additional input.

---

> > > ### Author Response · Authors · 2023-11-21
> > > **Response to Reviewer dXPD (Part 2)**
> > >
> > > II. Compute-cost/benefit trade-off. The training and sampling for a meaningful evaluation of each fusion requires expensive compute. For example, training the imagenet classifier from scratch for 500K steps with a batch size of 256 requires 46 V100 days as explained in Dhariwal and Nichol (2021b), and this does not consider any hyperparameter tuning. Our concern is that even if we were to carry out such an evaluation properly, these experiments (all of which would be added to the appendix) would not change our core contributions. This is because we already study in detail how using denoised example helps in the case of CE-loss. While we recognize that there will surely be *some* advantage by combining our approach with objectives which have improved over CE-loss, we feel that in order to present conclusions that are meaningful and reproducible, we would need to provide readers with a careful evaluation of “just how much” that fusion adds. To achieve that would require substantial tuning and repeated runs. That is, it would not change the fact that denoised examples are useful, but measuring exactly how useful they are, when combined with other techniques, we believe is outside the scope of this paper, whose focus is on introducing this new approach, providing theoretical motivation for it, and evaluating it in the *absence* of additional confounding elements.
> > >
> > > While we realize that this explanation does not provide the experiments that the reviewer suggested, we hope that we have clearly explained our reasons, and if any part of this argument is insufficient, we would be very glad to have the opportunity to discuss it further.
> > >
> > > [1] G. Zheng, S. Li, H. Wang, T. Yao, Y. Chen, S. Ding, and X. Li. Entropy-driven sampling and training scheme for conditional diffusion generation. ECCV, 2022.
> > >
> > > 2) Semi-supervised generation results. We are indeed working on the revision!

---

> > > ### Comment · Reviewer_dXPD · 2023-11-22
> > >
> > > Thank you for the detailed explanation. I understand that the comparison is need to be based on the same sampling method. It is beneficial to provide the DA-Classifier+EDS results.
> > >
> > > However, I'm still concerned about the orthogonality. In particular, the explanation of complementary is not sufficient because it does not guarantee that the expanded solution set has more good solutions than the original solution set. Moreover, this logic holds for any augmented classifier, not just for denoised examples. Therefore, I think this explanation is for the advantage of the additional input, but not for an advantage of the denoised assistant.

---

> > > > ### Author Response · Authors · 2023-11-23
> > > > **Respone to Reviewer dXPD (Part 3)**
> > > >
> > > > Thank you again for your response! Firstly, we thank you for acknowledging the benefits of our method over ED.
> > > >
> > > > Orthogonality: Thank you for reading our response and taking the time to raise your remaining concerns with the logic presented above. We hope that the following comments will provide a helpful perspective:
> > > >
> > > > 1) *“moreover, this logic holds for any augmented classifier, not just for denoised examples”.*
> > > >
> > > >     Absolutely– thank you for recognizing this! We agree that the logic holds for any classifier with additional inputs and is not specific for denoised examples – an important consequence of this observation is precisely that it does not prohibit training the DA-Classifier with novel loss objectives proposed in baselines. In other words, the DA-Classifier and the suggested baselines are not competing techniques!
> > > >
> > > > 2) *“the explanation of complementary is not sufficient because it does not guarantee that the expanded solution set has more good solutions than the original set”*
> > > >
> > > >    To be clear, our explanation is not intended to guarantee better solutions. Rather, our explanations guarantee that the solutions will not be worse, and our experiments show that the solutions are indeed better: Specifically, we demonstrate significant advantages of DA-classifiers over noisy-classifiers when trained on cross-entropy loss — i.e. the expanded solution set indeed has better solutions than the original set for the cross-entropy loss. Furthermore, we also showed that DA-classifiers trained with vanilla cross-entropy loss already perform better than noisy-classifiers trained with novel loss-objectives (e.g., DLSM/ECT).
> > > >
> > > > In conclusion, we should compare between classifiers trained on identical losses to determine the importance of denoised examples as additional inputs. We consider cross-entropy loss and extensively evaluate the advantages of denoised examples in classifier-guidance. Considering that the cross-entropy loss is still central to the novel loss objectives in baseline methods, we thus show there is potential in future work for improvements in training DA-Classifiers (instead of noisy classifiers) with the novel loss objectives proposed in baselines.
> > > > ***
> > > > We now address the remaining points from your earlier response.
> > > >
> > > > 3) Classifier-free Guidance in Semi-supervised Settings:
> > > >
> > > > To substantiate our explanations with experiments as requested, we trained a semi-supervised classifier-free guidance model on MNIST for 10 epochs. We construct the training dataset with 1k labeled images – by selecting 100 random examples per class – and 59k unlabeled images. We used a batch size of 1200 with 1k labeled images per batch and 200 unlabeled images – the labeled portion in each batch is approximately 83%, i.e. in accordance with the original recommendation in Ho and Salimans (2022). (Note that this setting is consistent with what we described in our previous, theoretical, response.) We trained it on VE-SDE using a U-Net having 16M parameters. For each epoch, we compute: (1) Train-Loss: the average score-matching loss measured at each training step, and (2) the average score-matching loss on validation Images (the 10k test-examples) at the end of each epoch.
> > > >
> > > > | Epoch | Train-Loss | Validation-Loss |
> > > > |:---:|:---:|:---:|
> > > > | 0 | 154.331894 | 64.599884 |
> > > > | 1 | 66.352173 | 58.647469 |
> > > > | 2 | 59.916943 | 58.456848 |
> > > > | 3 | 55.387962 | 63.058388 |
> > > > | 4 | 47.896942 | 93.815903 |
> > > > | 5 | 40.813591 | 124.892509 |
> > > > | 6 | 36.009224 | 153.456253 |
> > > > | 7 | 33.028023 | 163.888306 |
> > > > | 8 | 30.663017 | 199.100357 |
> > > > | 9 | 29.002604 | 203.914154 |
> > > >
> > > > We can see that the train-losses continue sharply decreasing while the validation-loss decreases for the first 3 epochs and then steadily increases. Thus, as expected based on our argument, we empirically observe classic overfitting in the implementation of semi-supervised classifier-free guidance. We hope that this experiment helps resolve your concerns about the validity of our reasoning.
> > > >
> > > > 4) Total derivative: Thank you for your clarification; we misunderstood the question asked in the original review. We analyze the total-derivative in Eq. (8) and visualize each component of the derivative in Fig. 7. We can answer “how the effect of a change in $\bf x$ is reflected in the denoised sample” by analyzing $\frac{\partial \hat{\bf x}}{\partial {\bf x}}$ – this is exactly what we investigate in Theorem 1. Is that the discussion that you are looking for? (We apologize if we still are not correctly understanding your question.)
> > > >
> > > > We hope this addresses all your concerns and look forward to resolving any outstanding concerns.

---

> > > > > ### Comment · Reviewer_dXPD · 2023-11-23
> > > > >
> > > > > Thank you for the response. I will consider the orthogonality discussion you mentioned along with the other reviewers.
> > > > >
> > > > > For 3, I want to know if the losses being compared are conditional or unconditional scores. Also, I think this comparison should include how the traditional CFG changes in train and validation loss.
> > > > >
> > > > > For 4, in a practical implementation, do we need to calculate each term on the right-hand side of Eq. 8? If not, I'm wondering how we can get the total derivative, which is why I was curious about the sampling procedure for this in the first review.

---

> > > > > > ### Author Response · Authors · 2023-11-23
> > > > > > **Response to Reviewer dXPD**
> > > > > >
> > > > > > Thank you so much for continuing to engage in the discussion.
> > > > > >
> > > > > > *For 3, I want to know if the losses being compared are conditional or unconditional scores. Also, I think this comparison should include how the traditional CFG changes in train and validation loss.*
> > > > > >
> > > > > > Every batch (train/test) has 1k labeled examples and 200 unlabeled examples. In traditional CFG, as expected, there will not be any discernible difference between train and validation losses.
> > > > > >
> > > > > > *For 4, in a practical implementation, do we need to calculate each term on the right-hand side of Eq. 8? If not, I’m wondering how we can get the total derivative, which is why I was curious about the sampling procedure for this in the first review.*
> > > > > >
> > > > > > Standard differentiation frameworks (e.g., autodiff) can automatically compute the total-derivative. We can find examples of total-derivatives in almost every neural architecture: e.g., feedforward neural networks, resnets, etc. As in every case, we have to only implement forward-pass and the backward pass automatically computes the total derivative.

---

> > > > > > > ### Comment · Reviewer_dXPD · 2023-11-23
> > > > > > >
> > > > > > > Thank you for your response. Then, is there a time difference due to different classifier gradient calculation between noisy classifier and DA classifier?

---

> > > > > > > > ### Author Response · Authors · 2023-11-23
> > > > > > > > **Response to Reviewer dXPD**
> > > > > > > >
> > > > > > > > Yes, there is a time difference. We have discussed this in the limitations section on page 12.

---

> > > > > > > > > ### Comment · Reviewer_dXPD · 2023-11-23
> > > > > > > > >
> > > > > > > > > Could you provide the actual sampling time between the noisy classifier and the DA classifier? Additionally, I have a concern that they suggest the distilling diffusion models would be a solution to solve this problem, but these methods are focused on the iterative sampling procedure, not for additional time due to classifier gradient.

---

> > > > > > > > > > ### Author Response · Authors · 2023-11-23
> > > > > > > > > > **Response to Reviewer dXPD**
> > > > > > > > > >
> > > > > > > > > > Thank you very much for your response!
> > > > > > > > > >
> > > > > > > > > > Upon acceptance, we will be glad to include systematic comparisons of sampling times between noisy classifier and DA classifier.
> > > > > > > > > >
> > > > > > > > > > In class-conditional generation, classifier-gradients and score-functions are evaluated in every step of the iterative sampling; recent research works explore distillation as an option for reducing the number of evaluations of score-functions in unconditional generation. Similarly, future research may explore distillation options for reducing number of evaluations in class-conditional generation and this could help reduce the sampling time. Improving sampling time in diffusion models is an active area of research and we suggest distillation as an example research direction that could potentially help improve running times.
> > > > > > > > > >
> > > > > > > > > > Finally, we also emphasize that the difference in running times can also be minimized by specialized engineering techniques. Thus, the comparison between wall-clock times should not take precedence over the main theoretical and empirical results of the paper.
> > > > > > > > > >
> > > > > > > > > > We hope that we have resolved your concerns.
> > > > > > > > > >
> > > > > > > > > > We also take this opportunity to sincerely thank you for your continued engagement with us throughout the discussion period!

---

### Author Response · Authors · 2023-11-23
**Rebuttal Revisions**

We thank all reviewers for their thoughtful and detailed feedback. We have uploaded a new revision of our paper incorporating the minor corrections and general revisions to the Introduction, DA-Classifier and Semi-supervised Training sections in order to provide more intuitions, improve readability, and further clarify our contributions. We also include additional results and baseline comparisons prompted through discussions with reviewer dXPD.

We believe that this has strengthened the paper, and we hope that we have addressed all the concerns.

---

### Meta-Review · Area_Chair_MkEL · 2023-12-14

**Metareview:**

The paper introduces Denoising-Assisted (DA) classifiers in diffusion models, aiming to improve both generative and discriminative performance, and includes semi-supervised diffusion framework applications.  Initially, Reviewer dXPD expressed concerns regarding baseline comparisons, missing generation results, CFG in semi-supervised settings, and clarity of the sampling procedure. The authors' rebuttal addressed some of these issues by providing additional baseline comparisons and clarifying the sampling procedure. dXPD was actively engaged in the discussion and followed up with more lengthy discussions with the author and required more systematic comparisons of sampling times between noisy classifier and DA classifier. In general, dXPD kept the initial rating and remained concerned about the orthogonality of the proposed method with existing techniques and the need for further empirical evidence in specific areas. Reviewer YWxN's concerns about numerical improvements and motivations for using diffusion samples in semi-supervised settings. Reviewer F98v's concerns about presentation and provided a final rating of marginal rejection. Lastly, Reviewer JmSU raised some concerns regarding section 4 of the paper.

Overall, the authors made commendable efforts to address the reviewers' concerns, leading to improvements in their paper. Despite this, the AC's decision leans towards rejection. The difference before and after the rebuttal is noticeable, but it wasn't sufficient enough to fully convince the reviewers and the AC of the paper's contribution. The paper clearly would benefit from another round of substantial revision.

**Justification For Why Not Higher Score:**

While the paper has merits and shows good improvement to diffusion classifiers, the remaining concerns from reviewers, particularly regarding the comparative effectiveness and clarity of contributions and messages, and especially considering the settings in the paper require more organization and prioritization, justify the decision not to recommend a higher score. A major revision would greatly enhance the potential impact of the work.

**Justification For Why Not Lower Score:**

N/A

---

### Decision · Program_Chairs · 2024-01-16

Reject